# "Who experiences large model decay and why?"
# A Hierarchical Framework for Diagnosing Heterogeneous Performance Drift

**Harvineet Singh** [1]   **Fan Xia** [1]   **Alexej Gossmann** [2]   **Andrew Chuang** [1]   **Julian C. Hong** [1]   **Jean Feng** [1]

## Abstract

Machine learning (ML) models frequently experience performance degradation when deployed in new contexts. Such degradation is rarely uniform: some subgroups may suffer large performance decay while others may not. Understanding where and how large differences in performance arise is critical for designing *targeted* corrective actions that mitigate decay for the most affected subgroups while minimizing any unintended effects. Current approaches do not provide such detailed insight, as they either (i) explain how *average* performance shifts arise or (ii) identify adversely affected subgroups without insight into how this occurred. To this end, we introduce a **S**ubgroup-scanning **H**ierarchical **I**nference **F**ramework for performance drif**T** (SHIFT). SHIFT first asks "Is there any subgroup with unacceptably large performance decay due to covariate/outcome shifts?" (*Where?*) and, if so, dives deeper to ask "Can we explain this using more detailed variable(subset)-specific shifts?" (*How?*). In real-world experiments, we find that SHIFT identifies interpretable subgroups affected by performance decay, and suggests targeted actions that effectively mitigate the decay.[1]

## 1. Introduction

ML algorithms are known to degrade in performance when applied in different contexts, which has led to extensive work on explaining how differences in an ML algorithm's *average* performance arise (Cai et al., 2023; Zhang et al., 2023). However, *performance differences are rarely uniform* in practice: some subgroups may experience severe performance degradation while others may experience very negligible differences, if at all (Yang et al., 2023). Understanding the subgroups where shifts are most pronounced and providing subgroup-specific explanations is critical from the perspectives of algorithmic fairness (Mitchell et al., 2021) and backwards compatibility (Srivastava et al., 2020). Subgroup-level explanations can also help model developers design *targeted* corrective actions that only modify the algorithm's behavior in the most affected subgroups and limit any other unintended effects ("Don't fix what ain't broke") (Globus-Harris et al., 2022; Suriyakumar et al., 2023).

For instance, suppose an ML algorithm for predicting unplanned readmission achieves overall accuracy of 85% in hospital A and 83% in hospital B. While the change in overall accuracy may not be clinically significant, the change within some subgroup may be sufficiently large to be deemed harmful. If so, it is natural to ask how this heterogeneity in performance decay arose: was it due to a change in how certain diseases are recorded, which medications are prescribed for certain patients, or something else altogether? If we know the affected subgroup and why, we can specifically address the root cause, such as by updating data pre-processing and/or the algorithm within the subgroup.

As such, our goal is to simultaneously understand *where* an ML algorithm performs substantially worse and *how* it arose. Numerous methods have been developed to find subgroups where an ML algorithm performs poorly (d'Eon et al., 2022; Eyuboglu et al., 2022; Liu et al., 2023; Subbaswamy et al., 2024), which can in principle be extended to identify subgroups with large model decay. Answering "how" is more tricky. We can obtain an approximate high-level answer by decomposing the average performance drop within an identified subgroup into the contribution from a shift in the marginal distribution of the input features $X$ (covariate shift) versus a shift in the conditional distribution of the target $Y|X$ (outcome shift) (Quinonero-Candela et al., 2009; Cai et al., 2023).

However, this is only a partial solution. For one, it misses situations where the subgroup of individuals experiencing severe covariate shifts is not the same as the one experiencing severe outcome shifts, and each subgroup may require different corrective actions. More importantly, we often

---

[1]University of California, San Francisco, USA [2]Independent researcher. Correspondence to: Jean Feng <jean.feng@ucsf.edu>.

*Proceedings of the 42nd International Conference on Machine Learning*, Vancouver, Canada. PMLR 267, 2025. Copyright 2025 by the author(s).

[1]Code is available at http://github.com/jjfeng/shift.

want to know precisely which subset of input variables were involved, as many real-world shifts involve only a few variables (i.e. sparse) and can be fixed in a targeted manner (Castro et al., 2020; Finlayson et al., 2021). Existing methods are currently insufficient, as they rely on assumptions that often do not hold in practice, e.g. the true causal graph is known (Zhang et al., 2023; Quintas-Martinez et al., 2024), the data follows simple parametric models (Baron & Kenny, 1986), or unrealistically large datasets (Singh et al., 2024).

To overcome these limitations, we present a nonparametric **S**ubgroup-scanning **H**ierarchical **I**nference **F**ramework for performance drif**T** (SHIFT) (Fig 1). Whereas prior works have approached drift diagnosis primarily through the lens of estimation, SHIFT approaches this through hypothesis testing. The advantage is that hypothesis tests answer simple yes/no questions, which is often more feasible in settings with limited data; in fact, we conduct *omnibus* tests, which require even less data as they do not need to identify the entire subgroup that is adversely affected. Furthermore, hypothesis tests allow us to check the very assumptions that other works have take on face value. The first stage of SHIFT performs a high-level analysis: decomposing distribution shift into an "aggregate" covariate shift with respect to all of $X$ and an "aggregate" outcome shift with respect to all of $X$, SHIFT tests if either have led to unacceptably worse performance in any meaningfully large subgroup (*Where?*). If so, the second stage drills down to test if this can be adequately explained by a shift solely with respect to a sparse subset of variables in $X$ (*How?*). The major contributions of this work are:

- Introduction of a novel hierarchical hypothesis testing framework that detects subgroups experiencing large performance decay due to aggregate-level covariate/outcome shifts, which are then explained using detailed variable(subset)-specific shifts.
- SHIFT does not rely on strong assumptions and is suitable for smaller datasets, making it broadly applicable to real-world scenarios.
- Our simulations demonstrate that SHIFT correctly identifies relevant shifts. Real-world experiments show that SHIFT can guide the design of model/data corrections that strictly improve performance.

## 2. Related Work

We briefly discuss the three most related areas below (summarized in Table 1). See Appendix E for more detailed discussion as well as other related areas.

**Detecting distribution shifts.** Many methods have been developed to detect *any* shift in marginal/conditional distributions, such as Kolmogorov-Smirnov (KS) (Rabanser et al., 2019), kernel-based tests (Zhang et al., 2011), and

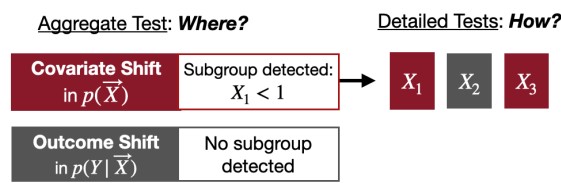

Figure 1: **S**ubgroup-scanning **H**ierarchical **I**nference **F**ramework for performance drif**T** (SHIFT) is a two-stage hypothesis testing procedure that first checks if there is a subgroup with unacceptably large performance decay due to aggregate covariate and outcome shifts with respect to all $X$ variables. If so, it checks if this can be explained by detailed variable(subset)-specific shifts. Red indicates the shift was flagged for further investigation. In this example, covariate shift is flagged because it affected a subgroup and variables $X_1$ and $X_3$ were flagged as potential explanations.

Maximum Mean Discrepancy (MMD) (Gretton et al., 2012a; Luedtke et al., 2018). More recent works focus on detecting only those that are harmful to *overall* performance (Podkopaev & Ramdas, 2022; Panda et al., 2024). No prior methods have been developed to specifically detect distribution shifts that lead to disproportionate harm in a (sufficiently large) subgroup, which SHIFT aims to address.

**Decomposing model performance.** Various methods have been developed to quantify the contribution of each feature subset to the *average* performance (Budhathoki et al., 2021; Cai et al., 2023; Wu et al., 2021; Zhang et al., 2023; Quintas-Martinez et al., 2024) and, more recently, the variability of performance changes (Singh et al., 2024). Mathematically, these methods rely on techniques similar to those used in mediation analysis for decomposing the average treatment effect into indirect and direct effects (Baron & Kenny, 1986) and variable importance (VI) methods for explaining the variability of the conditional average treatment effect (CATE) (Hines et al., 2023), respectively. Most of these methods either assume a parametric model or knowledge of the causal graph between individual variables. While VI methods that focus on decomposing variability have much weaker assumptions (Hines et al., 2023; Singh et al., 2024), they generally require large datasets and their confidence intervals (CIs) cannot be easily inverted to produce valid hypothesis tests (the influence function is degenerate because the estimand is at the boundary of the parameter space under the null, leading to inflated Type I error rates) (Hudson, 2023). Through carefully framed hypothesis tests, SHIFT provides valid statistical inference without parametric assumptions or knowledge of a detailed causal graph.

**Discovering subgroups.** Methods have been developed to identify subgroups with low performance within a single distribution (Eyuboglu et al., 2022; d'Eon et al., 2022; Ali et al., 2022; Feng et al., 2024a; Dong et al., 2024; Rauba

Table 1: **S**ubgroup-scanning **H**ierarchical **I**nference **F**ramework for performance drif**T** (SHIFT) compared to prior work

| Category (Example methods) | Detects subgroup with large decay? | Valid hypothesis test? | Avoids detailed causal graph? | Detailed explanations for outcome/covariate shifts? |
|---|---|---|---|---|
| Detect any shift (Rabanser et al., 2019; Zhang et al., 2011) | No | Yes | Yes | Outcome only |
| Detect loss shift (Podkopaev & Ramdas, 2022) | No | Yes | Yes | No |
| Decompose average perf decay (Zhang et al., 2023; Cai et al., 2023; Quintas-Martinez et al., 2024) | No | Some methods | No | Covariate only |
| Decompose shift variability (Singh et al., 2024) | No | No | Yes | Outcome & Covariate |
| Decompose ATE (Baron & Kenny, 1986) | No | Parametric only | No | Covariate only |
| Decompose CATE variability (Hines et al., 2023) | No | No | Yes | Outcome only |
| Subgroup discovery (Eyuboglu et al., 2022; d'Eon et al., 2022) | No | Some | Yes | No |
| **SHIFT** *(Proposed)* | Yes | Yes | Yes | Outcome & Covariate |

et al., 2024; Subbaswamy et al., 2024) and subgroups with large CATE (Athey et al., 2019). However, most methods only provide point estimates and not statistical inference (CIs/hypothesis tests). More importantly, no existing methods can be directly adapted to explain how large performance decay arises across subgroups with respect to variable-specific shifts.

## 3. Hierarchical Testing Framework

Given a set of features $X$ and an outcome $Y$, we want to understand the difference in performance of an algorithm $f$ across source and target domains, denoted by $d = 0$ and $d = 1$ respectively. We refer to the joint distribution of $(X, Y)$ in each domain by $p_d$ and its corresponding expectation with $E_d$. Performance is quantified by a loss function $\ell := \ell(y, f(x)) \in \mathbb{R}$. The average loss conditional on $x$ in domain $d$ is denoted $Z_d(x) := E_d[\ell(Y, f(X))|X = x]$ for $d \in \{0, 1\}$. Hat notation denotes an estimate.

A shift in the joint distribution of $(X, Y)$ can be decomposed into aggregate covariate and outcome shifts, which are defined by the shifts $p_0(x) \Rightarrow p_1(x)$ and $p_0(y|x) \Rightarrow p_1(y|x)$, respectively. In this way, the shift from source to target can be broken down into a sequence of aggregate-level shifts:

$$p_0(x)p_0(y|x) \Rightarrow p_1(x)p_0(y|x) \Rightarrow p_1(x)p_1(y|x) \quad (1)$$

and, correspondingly, the average performance change can be decomposed into:

$$E_1[\ell] - E_0[\ell] = \underbrace{E_1[Z_0] - E_0[Z_0]}_{\text{covariate shift}} + \underbrace{E_1[Z_1 - Z_0]}_{\text{outcome shift}} \quad (2)$$

To generate even more detailed explanations of performance shifts, we will consider sparse shifts solely with respect to variable subsets $X_s$ and use $p_s(x)$ and $p_s(y|x)$ to denote $X_s$-specific covariate and outcome shifts, respectively. We will present their exact definitions later.

SHIFT is a hierarchical diagnostic framework that does a more detailed analysis of performance drift compared to the standard two-way decomposition in (2) by accounting for heterogeneity of performance shifts. At the first level, SHIFT checks if the aggregate covariate and outcome shifts lead to subgroups with large performance decay. If so, SHIFT searches for a more detailed explanation among candidate variable(subset)-specific shifts.

Throughout, SHIFT focuses only on subgroups of individuals and performance shifts that are deemed large enough to be of practical interest, by requiring the domain expert to select a priori the minimum subgroup size $\epsilon > 0$ and minimum shift magnitude $\tau \geq 0$. This is critical to ensure the practical usability of these methods, as alarms for negligible shifts lead to alarm fatigue (Cvach, 2012; Feng et al., 2025). The set $\mathcal{A}_\epsilon$ denotes all subgroups whose prevalence in the source and target domains exceed $\epsilon > 0$.

The following two sections (Sec 3.1 and 3.2) introduce the aggregate and detailed hypothesis tests in SHIFT and Section 4 describes the actual testing procedures.

### 3.1. Aggregate tests: *Where?*

SHIFT first tests if there exists a subgroup with large performance decay due to an aggregate covariate shift and, likewise, a subgroup impacted by an aggregate outcome shift. The impacts of these shifts within a subgroup $A \in \mathcal{A}_\epsilon$

is quantified using a similar decomposition as (2), i.e.

$$E_1[\ell|X \in A] - E_0[\ell|X \in A]$$
$$= \underbrace{E_1[Z_1 - Z_0|X \in A]}_{\text{outcome shift}} + \underbrace{E_1[Z_0|X \in A] - E_0[Z_0|X \in A]}_{\text{covariate shift}}.$$

This leads to tests for the following null hypotheses:

---

**Hypothesis 3.1** (Agg covariate shift). $H_0^X$: *For all subgroups $A \in \mathcal{A}_\epsilon$, the performance drift in $A$ due to the aggregate covariate shift is no larger than tolerance $\tau \geq 0$, i.e. $E_1[Z_0(X)|X \in A] - E_0[Z_0(X)|X \in A] \leq \tau$.*

---

**Hypothesis 3.2** (Agg outcome shift). $H_0^{Y|X}$: *For all subgroups $A \in \mathcal{A}_\epsilon$, the performance drift in $A$ due to the aggregate outcome shift is no larger than tolerance $\tau \geq 0$, i.e. $E_1[Z_1(X) - Z_0(X)|X \in A] \leq \tau$.*

---

For each shift mechanism, rejection of the null means that there is a subgroup of concern and further investigation is warranted, thereby triggering a second stage of testing. Before diving into the second stage, we discuss connections between these aggregate tests and the existing literature.

**Connection to MMD.** The proposed tests assess for distributional differences by comparing the maximum difference in the mean loss along the shift sequence in (1). This shares similarities to MMD, which also measures the distance between two distributions in terms of the maximum difference in expected value over some function class (often referred to as the "critic") (Gretton et al., 2012a). To see the connection more formally, we rewrite the above tests in terms of binary detectors where $h_A(X) = \mathbb{1}\{X \in A\}$ for subgroup $A$. Define the critic function class to be the set of "filtered" loss functions $\{(x, y) \mapsto h_A(X)\ell(f(x), y) : A \in \mathcal{A}_\epsilon\}$. For the first two distributions in (1), MMD defines their distance as the maximum average difference of the filtered loss, i.e.

$$\sup_{A \in \mathcal{A}_\epsilon} E_{10}[\ell(X,Y)h_A(X)] - E_{00}[\ell(X,Y)h_A(X)],$$

where $E_{d_1,d_2}$ indicates the expectation with respect to distribution $p_{d_1}(X)p_{d_2}(Y|X)$. In contrast, the aggregate covariate shift test can be viewed as measuring the maximum average difference of the *conditional* loss, i.e.

$$\sup_{A \in \mathcal{A}_\epsilon} \frac{E_{10}[\ell(X,Y)h_A(X)]}{E_{10}[h_A(X)]} - \frac{E_{00}[\ell(X,Y)h_A(X)]}{E_{00}[h_A(X)]}.$$

A similar analogy holds for the aggregate outcome shift, which compares the last two distributions in (1). Thus, SHIFT can be viewed as testing the Maximum *conditional*-Mean Discrepancy (McMD) rather than the MMD. Like MMD, McMD is zero when the compared distributions are equal. Unlike MMD, McMD can be large even when the mean difference is large in only a small subgroup, reflecting its priority placed on algorithmic fairness.

**Connection to mediation analysis.** Prior works have highlighted that the decomposition of *average* performance change into covariate and outcome shifts parallels the decomposition of the *average* treatment effect into indirect and direct effects, which is commonly analyzed in causal mediation analysis (Castro et al., 2020; Singh et al., 2024). As this work decomposes *subgroup-specific* performance changes, it parallels recent efforts in the nascent but growing field on analyzing the heterogeneity of causal effect decompositions (Loh et al., 2020; Rubinstein et al., 2023). The omnibus tests developed in this work may thus be useful for testing heterogeneous indirect/direct effects, an area that has not been addressed thus far. We discuss these connections further in Appendix A.

### 3.2. Detailed tests: *How?*

For each shift mechanism, rejection of the first-stage test implies that there is a subgroup for which performance change was large. The next step is to find a detailed explanation, by identifying the variables most likely to be responsible.

SHIFT finds explanations by searching over a suite of candidate shifts with respect to individual variables or variable subsets. Because the true causal graph is not typically known in practice, the set of all possible variable(subset)-specific shifts is exponentially large and a comprehensive search over all such shifts is computationally intractable. As such, SHIFT considers a restricted set of detailed candidate shifts as potential explanations. In this work, given a variable subset $X_s$, we consider the following:

- **Outcome shift**: We consider the candidate $p_s(y|x) := p_1(y|x_s, \mu_0(x))$, where $\mu_0(x) = p_0(y = 1|x)$ is the outcome probability at the source. This is similar to shifts considered in model recalibration (Steyerberg, 2009), where the shift is defined relative to the outcome's original conditional probability in the source domain.
- **Covariate shift**: We consider the candidate $p_s(x) := p_1(x_s)p_0(x_{-s}|x_s)$. Such a shift may occur, for instance, if $X_s$ precedes $X_{-s}$ causally and is commonly considered in prior works (Wu et al., 2021; Zhang et al., 2023; Singh et al., 2024).

Other candidate shifts are certainly possible (see Sec F) and we leave them to future work. Critically, unlike prior works that offer variable-level explanations of performance decay assuming these candidate shifts are actually true (Wu et al., 2021; Zhang et al., 2023), SHIFT does *not* assume that these candidate shifts are correctly specified because everything is conducted through the lens of hypothesis testing. Instead, SHIFT tests whether a candidate offers a good explanation.

Given candidate shifts, we now quantify how well they explain the heterogeneous performance changes in the data. We say that an aggregate covariate shift is well-explained

by an $X_s$-specific covariate shift if the performance change induced by the former is well-approximated by the latter across all subgroups $A$, i.e. for all $A \in \mathcal{A}_\epsilon$,

$$E_1[Z_0|X \in A] - E_0[Z_0|X \in A]$$
$$\approx E_s[Z_0|X \in A] - E_0[Z_0|X \in A],$$

where $E_s(X)$ is with respect to an $X_s$-specific covariate shift. Similarly, an aggregate outcome shift is well-explained by an $X_s$-specific outcome shift if for all $A \in \mathcal{A}_\epsilon$,

$$E_1[Z_1 - Z_0|X \in A] \approx E_1[Z_s - Z_0|X \in A], \quad (3)$$

where $Z_s(X)$ is the expected loss under the candidate shift. This is formalized in detailed tests of $X_s$-specific shifts with the following null hypotheses:

> **Hypothesis 3.3** ($X_s$-specific covariate shift). $H_{0,s}^X$: *For all subgroups $A \in \mathcal{A}_\epsilon$ and tolerance $\tau$, the candidate $X_s$-specific covariate shift explains the performance change, i.e., $E_1[Z_0(X)|X \in A] - E_s[Z_0(X)|X \in A] \leq \tau$.*

> **Hypothesis 3.4** ($X_s$-specific outcome shift). $H_{0,s}^{Y|X}$: *For all subgroups $A \in \mathcal{A}_\epsilon$ and tolerance $\tau$, the candidate $X_s$-specific outcome shift explains the performance change, i.e., $E_1[Z_1(X) - Z_s(X)|X \in A] \leq \tau$.*

If we fail to reject the null for an $X_s$-specific covariate or outcome shift, SHIFT flags it as potentially important. Then for some prespecified $\alpha > 0$, the potentially important variable subsets for covariate and outcome shifts are

$$\hat{\mathcal{S}}_n^{\texttt{shift}} = \left\{ s : p\text{-value for } H_{0,s}^{\texttt{shift}} > \alpha \right\} \quad (4)$$

for $\texttt{shift} = Y|X$ and $\texttt{shift} = X$. A human expert can then verify which variables in $\hat{\mathcal{S}}_n^{\texttt{shift}}$ are the true root cause(s) and design targeted corrective actions.

Comparing the detailed and aggregate-level tests, one may notice that they have nearly the same mathematical structure and yet are interpreted differently to answer differing questions (*where?* versus *how?*). To see how this is possible, note that the tests could have been interpreted in the same way: aggregate-level tests check whether aggregate shifts are well-approximated by the zero function, i.e. whether $E_1[Z_1 - Z_0|X \in A] \approx 0$ and $E_1[Z_0|X \in A] - E_0[Z_0|X \in A] \approx 0$, while the detailed tests check if aggregate shifts are well-approximated by candidate $X_s$-specific shifts.

*Remark* 3.1 (Modified covariate shift tests). When we have features that are independent of the loss function, covariate shifts in such features may still be flagged which is undesirable. This occurs due to *collider bias* since conditioning on the subgroup $\mathbb{1}\{x \in A\}$ induces a correlation between the independent features and the loss function. Section B gives more details. As a remedy, we filter features that are uncorrelated with the loss function as a data preprocessing step and then run the covariate shift tests as usual.

### 3.3. Visualization of SHIFT

Results from SHIFT are visualized in a hierarchical plot (Fig 1), where "red" means "flagged" and "gray" means "not flagged." At the aggregate level, the covariate/outcome shift is "flagged" if a subgroup was found to have large performance decay due to that shift mechanism (null was rejected). To interpret aggregate-level test results, we summarize the detected subgroup using rule-based decision sets (Lakkaraju et al., 2016), although other ML explainability methods can be used instead. At the detailed level, we flag variable(subset)-specific covariate/outcome shifts that may offer a potential explanation of the heterogeneous performance shifts (null was not rejected). "Flag strength" is one minus the p-value for aggregate-level tests and the p-value for detailed tests. Note that if none of the candidate sparse shifts are adequate explanations, one may need to explore alternative shift explanations (e.g. less sparse).

## 4. Inference Procedure

We now describe the inference procedure for tests introduced in the previous section. We begin with rewriting each hypothesis test in terms of a simple target of inference. This will illuminate the general approach we would like to take, as well as the technical challenges we will encounter.

To illustrate, note that the aggregate outcome test can be equivalently expressed as testing the null hypothesis

$$H_0^{Y|X} : \underbrace{\sup_{A \in \mathcal{A}_\epsilon} E_1\left[(Z_1(X) - Z_0(X) - \tau)h_A(X)\right] \leq 0.}_{\text{target of inference}} \quad (5)$$

The target of inference can be interpreted as follows: $h_A$ is scaled by how much the difference in expected loss exceeds tolerance $\tau$, so the target of inference can be interpreted as the *Maximum Expected Exceedence* (MEE) between the last two distributions in the shift sequence in (1). Similarly, the detailed outcome test can be rewritten as

$$H_{0,s}^{Y|X} : \sup_{A \in \mathcal{A}_\epsilon} E_1\left[(Z_1(X) - Z_s(X) - \tau)h_A(X)\right] \leq 0. \quad (6)$$

The aggregate and detailed covariate tests can be interpreted similarly, though the scaling term is not as clean:

$$H_0^X : \sup_{A \in \mathcal{A}_\epsilon} E_0\left[(Z_0(X)(\tilde{\pi}_A(X) - 1) - \tau)h_A(X)\right] \leq 0 \quad (7)$$

$$H_{0,s}^X : \sup_{h \in \mathcal{A}_\epsilon} E_0\left[(Z_0(X)(\tilde{\pi}_A(X) - \tilde{\pi}_{s,A}(X)) - \tau)h_A(X)\right] \leq 0 \quad (8)$$

where $\tilde{\pi}_A(x) = \frac{p_1(x)E_0[h_A(X)]}{p_0(x)E_1[h_A(X)]}$ and $\tilde{\pi}_{s,A}(x) = \frac{p_1(x_s)E_0[h_A(X)]}{p_0(x_s)E_s[h_A(X)]}$ are scaled density ratios. Given this rewriting of the MEE, we can now discuss two technical challenges that we can resolve, in part, through *sample splitting*.

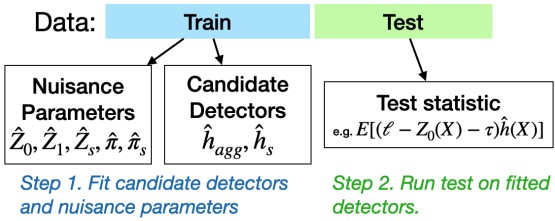

Figure 2: Overview of testing procedure

First, estimating a supremum over the infinite number of binary detectors $h_A$ is computationally intractable. Nevertheless, our goal is simply hypothesis testing, not estimation. We can accomplish this by sample splitting, where one portion of the data is for learning one (or a few) good candidate detector ($\hat{h}_A$) and the remaining data is for testing the expected exceedence for $\hat{h}_A$. This can be viewed as running a restricted version of the original test, where we only test the MEE with respect to the singleton set $\{\hat{h}_A\}$ rather than all of $\mathcal{A}_\epsilon$. While this approach may be conservative, it provides statistical guarantees with fewer assumptions and better finite sample behavior. The remaining question is how we can find good candidate detectors.

Second, the MEE involves unknown outcome models ($Z_d$) and scaled density ratio models ($\tilde{\pi}_A$ and $\tilde{\pi}_{s,A}$), which we collectively refer to as nuisance parameters. Prior works have shown that plug-in estimators, which use the same data to both train nuisance parameters and estimate targets of inference, are biased. Following results in double-debiased ML and semiparametric theory (Chernozhukov et al., 2018), we use sample splitting to remove some of this bias. The question is then how to remove the remaining bias for achieving the desired Type I error control.

Given the benefits of sample-splitting, the overall testing procedure uses this as the basis: *Step 1* estimates candidate detectors and nuisance parameters on a training partition and *Step 2* uses the remaining data to conduct a restricted test with respect to the fitted models (Figure 2). For ease of exposition, we describe the procedure for a single sample-split, but it can be easily extended with cross-fitting to improve statistical efficiency (Kennedy, 2024). Here we describe each step broadly and highlight key innovations needed to address the technical challenges. The detailed testing procedure (including hyperparameter selection) is given in Section C of the Appendix.

**Step 1. Estimate candidate detectors and nuisance parameters using the training partition.** The nuisance parameters can be estimated using ML following standard recipes (Kennedy, 2024). Estimating candidate detectors for the aggregate and detailed outcome shift tests is also straightforward. For the aggregate version, the estimand in (5) is maximized when the conditional mean

$E_1[(Z_1(X) - Z_0(X) - \tau)h_A(X)|X]$ is maximized, so the optimal detector is $h_A(X) = \mathbb{1}\{Z_1(X) - Z_0(X) - \tau > 0\}$. Consequently, we can take a plug-in approach to construct a candidate detector, i.e. $\hat{h}_A(X) = \mathbb{1}\{\hat{Z}_1(X) - \hat{Z}_0(X) - \tau > 0\}$. A similar approach can be taken for the detailed version.

Estimating candidate detectors for the covariate shift tests is, however, not immediately obvious. For instance, the MEE in (7) cannot be maximized by individually maximizing its conditional mean, because of the shared ratio term $E_0[h_A(X)]/E_1[h_A(X)]$. Instead, we find the optimal detector by solving *the dual for a sequence of optimization problems*. That is, we can reframe the task as solving

$$\sup_A E_0\left[\left(\hat{Z}_0(X)(\hat{\pi}(X)\omega - 1) - \tau\right)h_A(X)\right] \leq 0 \tag{9}$$
$$\text{s.t. } \omega = E_0[h_A(X)]/E_1[h_A(X)]$$

for some $\omega > 0$. Using the method of Lagrange multipliers, the solution must have the form $\hat{h}_A^{(\omega,\lambda)}(X) = \mathbb{1}\left\{\left(\hat{Z}_0(X) - \lambda\right)(\hat{\pi}(X)\omega - 1) \geq 0\right\}$ for some $\lambda \geq 0$. Thus we can estimate the optimal candidate detector by sweeping over a grid of $\omega$ and $\lambda$ values. We can estimate detectors for detailed covariate shifts in a similar manner.

**Step 2. Conduct double-debiased tests on held-out data.** On the remaining data, we construct asymptotically linear estimators for the MEE with respect to the fitted candidate detector(s), using the approach of one-step correction.[2] This is relatively straightforward for (5), (7), and (8) by noting the mathematical similarities between MEE and direct/indirect effects in causal mediation analysis. However, one-step correction for the detailed outcome shift does not follow from standard recipes, which require the target of inference to be pathwise differentiable. The problem is that (6) involves $Z_s(X)$, which is not pathwise differentiable because its definition involves indicator functions. Still, we can sidestep this issue by leveraging the binning trick in Singh et al. (2024). Rather than defining an outcome shift as a function of $\mu_0(x)$, we define a *binned* outcome shift that replaces all occurences of $\mu_0$ with a *binned* version. Assuming that the set of observations that fall exactly on the bin edges have measure zero, we can show that the MEE with respect to the binned outcome shift is pathwise differentiable, so to allow construction of an asymptotically linear estimator.

**Theoretical properties.** Under the assumptions described in Appendix D, we can prove that the estimators for the MEE with respect to fitted detectors are asymptotically linear and their respective tests control the Type I error and have power one, asymptotically. Consequently, for outcome and

---

[2]Appendix C.3 discusses a more statistically efficient but more complex procedure involving the Maximum conditional Expectation of the Exceedence (McEE) rather than the MEE. We discuss testing of the MEE in the main manuscript for ease of exposition.

covariate shifts (shift $= Y|X$ and shift $= X$), if there is a candidate detailed shift with respect to variable subset $s^{*,\text{shift}}$ that corresponds to the true shift, it will be flagged by SHIFT, i.e. $P(s^{*,\text{shift}} \notin \mathcal{S}_n^{\text{shift}}) \leq \alpha$.

# 5. Results

We now validate SHIFT in simulation studies where the ground truth is known and two real-world case studies. For comprehensive validation, we vary the type and degree of shifts, the ML algorithms under study, and the data sizes. We present a summary of the results here due to space constraints and provide full experiment details in Section G.

**SHIFT.** For all experiments, performance is defined in terms of the 0-1 misclassification loss. We fit ML models (e.g. gradient boosting trees (GBT)) for the nuisance parameters and detectors, with hyperparameters chosen through cross validation. The significance level is set to $\alpha = 0.05$.

**Baseline methods.** There is no existing comparator that provides universal testing for all four types of shifts (aggregate/detailed and covariate/outcome) for the exact formulations used in SHIFT. Given these constraints, different comparators are used for different shift types and, when necessary, adapted to be as close as possible.

For *aggregate* shifts, we compare against Kernel independence tests KCI (Zhang et al., 2011) and MMD (Gretton et al., 2012b). For *detailed outcome* shifts, we compare against (a) TE-VIM (Hines et al., 2023) which quantifies VI for explaining conditional average treatment effect, (b) ParamY which fits a parametric regression model of the outcome $Y$ given domain $D$, features $X$, and interaction terms $DX$ and determines VI based on coefficients of the interaction terms, (c) ParamLoss which is the same as ParamY except it regresses loss $\ell$, and (d) KCI (Zhang et al., 2011) which is a kernel conditional independence test for $D \perp \ell|X_s$. For *detailed covariate* shifts, we compare against (a) KS which is the classic Kolmogorov-Smirnov test for comparing two univariate distributions, (b) Score (Kulinski et al., 2020) which detects shifts in $X_s|X_{-s}$ via the Fisher score, and (c) KCI (Zhang et al., 2011) which is a kernel conditional independence test for $D \perp X_{-s}|X_s$.

## 5.1. Simulations

Here we illustrate how SHIFT is more powerful and identifies only relevant shifts, i.e. those that contribute to performance drifts of magnitude $\geq \tau$ in some subgroup with prevalence $\geq \epsilon$.

**Data generating process.** We generate variables $X$ from a multivariate normal distribution centered at $m_d$ and covariance $\Sigma_d$ for domain $d$ and binary outcome $Y$ per logit $\phi_d(x)$. The ML algorithm is a logistic regression model

fitted to data from the source domain. We take $n = 8000$ points from both source and target domains, and split them into halves for training and evaluation.

**Setup 1a/b** (Compare agg-level outcome/covariate tests): For $X \in \mathbb{R}^{10}$, the shift only occurs in subgroup $A = \{x|x_1 \notin [-3.5, 3.5]\}$. Setup 1a only shifts the outcome logits per $\phi_1(x) = \phi_0(x) - 0.6x_1 \mathbb{1}\{x \in A\}$; Setup 1b only shifts the mean of the first covariate. To make the tests comparable, SHIFT tests for $\tau = 0, \epsilon = 0.05$.

**Setup 2** (Compare detailed outcome test): $\phi_0(x) = 0.8x_1 + 0.5x_2 + x_3 + 0.6x_4$ and $\phi_1(x) = 0.2x_1 + 0.4x_2 + x_3 + 0.6x_4$. The outcome shifts with respect to both $X_1$ and $X_2$, but the shift in $X_2$ is minimal and below tolerance $\tau$. Accuracy drops by 5.9%. SHIFT tests for $\tau = 0.05, \epsilon = 0.05$.

**Setup 3** (Compare detailed covariate test): $m_0 = (1, 0, 0, 1), \Sigma_0 = \text{diag}(2, 2, 2, 2)$ and $m_1 = (0, 0, 0, 0), \Sigma_1 = \text{diag}(1, 2, 2, 2)$. Both $X_1$ and $X_4$ shift but $X_4$'s shift is very small and below tolerance $\tau$. Accuracy drops by 5.4%. SHIFT tests for $\tau = 0.02, \epsilon = 0.05$.

**SHIFT correctly identifies relevant shifts, achieves nominal type-I error rate, and is consistent.** In Setups 1a/b, the aggregate-level tests in SHIFT are considerably more powerful than KCI and MMD, which are both kernel-based methods that tend to do poorly in high dimensions (Table 2). In contrast, SHIFT takes advantage of flexible ML estimators, which allows it to recover the true subgroup $A$ with reasonable accuracy (73.7% and 41.9% in setups 1a and 1b, respectively). In Setups 2 and 3, the aggregate-level tests in SHIFT also correctly flag outcome shifts (Fig 3a) and covariate shifts (Fig 3b), respectively. At the detailed level, SHIFT correctly flags variable $X_1$ as being a good explanation for the large performance shifts; the others are ignored because they either do not contribute or have negligible impacts. In Appendix I, we also show that SHIFT controls the Type-I error rate and is consistent (asymptotically power-one).

Table 2: **Aggregate tests.** Power for detecting outcome or covariate shifts in a subgroup. Power is computed as the rejection rate among 25 random draws of the dataset. We observe that SHIFT has the highest power.

| Setup | SHIFT | KCI | MMD |
|---|---|---|---|
| 1a Outcome | **0.56** (0.42,0.7) | 0.26 (0.16,0.4) | 0.06 (0.02,0.16) |
| 1b Covariate | **0.94** (0.84,0.98) | 0.0 (0.0,0.0) | 0.46 (0.32,0.6) |

**Comparators do not flag the correct shifts.** For comparators in the *detailed outcome* test (Setup 2), we find the following: TE-VIM does not find any variable able to explain heterogeneity of performance drift because it has weird behavior at the null. KCI can only check if the "marginal"

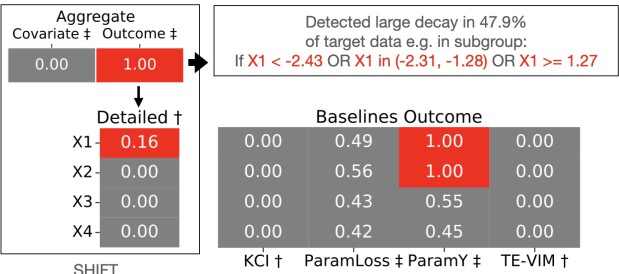

(a) **Setup 2**, only $X_1$-specific outcome shift should be flagged

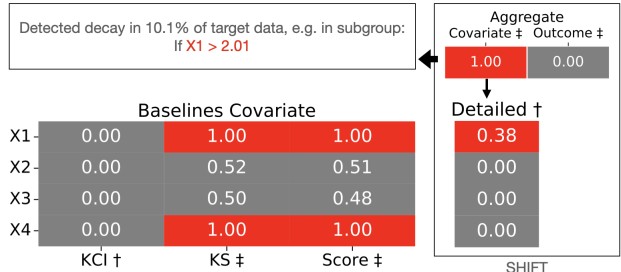

(b) **Setup 3**, only $X_1$-specific covariate shift should be flagged

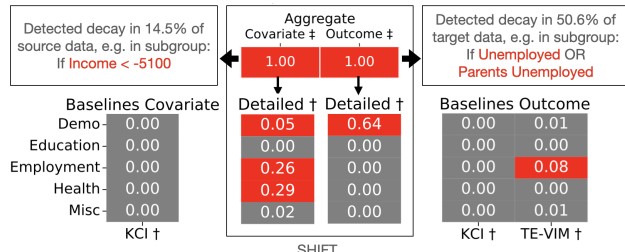

(c) **Insurance coverage** prediction with 5 variable subsets

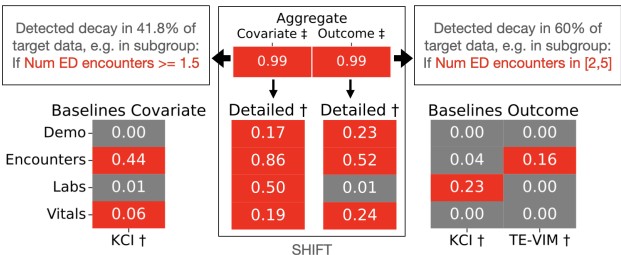

(d) **Readmission** prediction with 4 variable subsets

Figure 3: **Hypothesis testing results for variable(subset)-specific shifts.** SHIFT shown in outlined boxes; baselines for covariate and outcome shifts shown on the bottom left and right, respectively. Null hypotheses either state that a shift should be flagged (†), in which case we flag it in red if the p-value > 0.05 and show the p-value in the colored box, or that a shift should *not* be flagged (‡), in which case we flag it if the p-value ≤ 0.05 and show 1− p-value. For synthetic Setups 2 and 3, we report median p-values over 50 randomly-sampled datasets.

distribution of the loss can be explained by individual variables, i.e. if the loss distribution is independent of $D$ given $X_j$. This is a very specific type of explanation and does not hold, and so KCI fails to find any good explanation. ParamLoss is an incorrectly specified model and thus incorrectly flags none of the variables. ParamY is correctly specified model so it flags $X_1$ and $X_2$ as shifting, which is correct though does not respect the specified tolerance. Similar issues are found in the *detailed covariate* test (Setup 3). KCI, KS, and Score all incorrectly flag both $X_1$ and $X_4$ even though $X_4$ is irrelevant. This is because they check if $X_4$ has shifted, but do not account for the fact that $X_4$ is not actually used by the model nor does it affect $Y$ in any capacity.

### 5.2. Real-world case studies

**Health insurance prediction across states.** We study performance drift of an MLP trained to predict public health insurance coverage using census data from Nebraska, which is subsequently applied to Louisiana. Datasets have 3166 and 12000 points respectively and 34 features. Accuracy drops by 13.7% on average. At the aggregate-level, SHIFT finds that both outcome and covariate shifts affect subgroup-level accuracy (Fig 3c). For example, accuracy for the subgroup detected by the aggregate outcome test, comprising 50.6% of the target data, decays by 19.4%. Grouping the variables (34 in total) into 5 broad categories, we find from the detailed tests from SHIFT that shifts with respect to demographics can explain subgroup-level decay due to both shift types. Similar to that in simulations, the KCI baseline method struggles to find a good explanation while TE-VIM only flags employment-related variables. Based on these findings, we compare three ways to fix the model: a standard non-targeted (Non-T) fix that retrains the model for everyone with respect to all variables; a fix that retrains the model for everyone with respect to only the employment-related variables identified by TE-VIM; and a very targeted fix that only updates the model for the subgroup and the demographic variables detected by SHIFT (Table 3). We find that the targeted fix does better than the non-targeted fixes, and the non-targeted fixes inadvertently decay performance in other subgroups.

**Readmission prediction across hospitals.** The clinical AI field has developed numerous models to predict whether patients will be have an unplanned readmission after discharge from a hospital, which can be used to allocate extra resources to high-risk patients. We study a GBT readmission model trained on data from an academic hospital and transferred to a safety-net hospital. Since the hospitals serve different populations, the goal is to understand which exact shifts contributed the most to accuracy changes, such as changes in how patient variables are measured or changes in how care is delivered. Datasets from the academic and

Table 3: **Comparing model updates on insurance study.** We report AUC and 95% CI for performance of the original model and targeted versus non-targeted model updates, as measured with respect to the overall population (left column) and the subgroups where the original model (`Org`) and non-targeted model updates (`Non-T` and `TE-VIM`) experience large performance decays (right three columns). The targeted update based on SHIFT results performs better along all dimensions.

| Model | Overall | Subgroup for models | | |
| | | `Org` | `Non-T` | `TE-VIM` |
|---|---|---|---|---|
| Original model, `Org` | 69.2 (67.0,71.4) | 59.0 (55.3,62.3) | — | — |
| Non-targeted update, `Non-T` | 73.0 (71.0,75.0) | 67.3 (64.0,70.4) | 41.8 (24.6,59.2) | — |
| Update as per `TE-VIM` feats. | 73.0 (71.0,75.1) | 64.9 (61.6,67.9) | — | 63.7 (56.6,70.5) |
| Targeted update as per `SHIFT` | **74.8** (72.8,76.8) | **67.7** (64.4,70.4) | **66.4** (46.8,83.6) | **65.0** (57.5,71.5) |

safety-net hospitals have 7468 and 6515 points, respectively, and 27 features. Accuracy on average decays by 6.1% when the model is transferred. SHIFT detects significant changes in subgroup-level accuracy due to both aggregate outcome and covariate shifts (Figure 3d). For instance, the subgroup detected by the aggregate covariate test (comprising 41.8% of target data) has a 15.4% drop in accuracy. We find that the top feature highlighted by SHIFT for both covariate and outcome shifts is `num ED encounters`. When the same variable is highlighted for both covariate and outcome shifts, it can indicate that the definition of the variable has shifted. Investigating the data extraction procedure further, we indeed find this to be the case: the encounters feature was extracted differently across the hospitals. After correcting the extraction of this feature, covariate shifts no longer lead to a significant subgroup-level accuracy drop (p-value for the aggregate covariate test is no longer significant). This illustrates how SHIFT can help bridge accuracy gaps.

**Application to unstructured and high-dimensional data.** Although SHIFT is primarily designed for tabular data, its aggregate-level tests are suitable for analyzing unstructured data; its detailed-level tests can also be used, if one has prespecified concepts (Koh et al., 2020). As an example, we apply SHIFT to the CivilComments dataset (Koh et al., 2021), which contains comments on online articles and are judged to be toxic or not. Given 768-dimensional embeddings of the comments, SHIFT detects accuracy drops, as described in Section J.

## 6. Conclusion

We propose hypothesis tests to identify subgroups where an ML model decays in performance due to distribution shift across two contexts. The tests can also explain how the decay arises by checking for variable subset-specific shifts that can explain the decay. The tests can be configured to detect only meaningfully large performance decay and can be implemented readily using off-the-shelf ML models. Despite using ML estimators, we show that the tests have controlled false detection rate and good power asymptotically. Although the experiments here primarily focus on tabular data, SHIFT can be extended to unstructured data such as images and text by featurizing such data into concepts (Koh et al., 2020; Feng et al., 2024b). Our explorations with text data show that SHIFT provides a solid theoretical foundation on which future work can build.

## Acknowledgements

We would like to thank Lucas Zier, Patrick Vossler, Avni Kothari, and Romain Pirracchio for their helpful input and comments on this work. We are especially grateful to Adarsh Subbaswamy, Nicholas Petrick, and Gene Pennello, who provided invaluable feedback on the project from its inception to completion. We thank them for their tireless commitment. This work was funded through a Patient-Centered Outcomes Research Institute® (PCORI®) Award (ME-2022C125619). The views presented in this work are solely the responsibility of the author(s) and do not necessarily represent the views of the PCORI®, its Board of Governors or Methodology Committee, and the Food and Drug Administration. JCH acknowledges support from the National Cancer Institute of the National Institutes of Health (R01CA277782), which had no role in study design, data collection and analysis, decision to publish, or preparation of the manuscript.

## Impact Statement

The methods in the work can identify subgroups that experience overly large performance decay when an ML algorithm is transferred across domains or used over time. Results from SHIFT can be used to suggest interventions that can improve the impacted subgroup's performance, without substantively impacting other subgroups. We recommend working with domain experts to define what constitutes a meaningfully large subgroup and performance decay, as these determine what the test will aim to detect. These thresholds impact the interpretation of the tests and subsequent actions taken for closing performance drops.

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

# Appendix

## Table of contents

Table 4: **Summary of notation.**

| Notation | Meaning |
|---|---|
| $X, Y, f(X)$ | Covariates, Outcome, ML algorithm's prediction |
| $X_s$ | Subset of covariates (or variables) |
| $D \in \{0,1\}$ | Domain or dataset, 0 means source and 1 means target domain |
| $p_d(X,Y)$ | Probability density (also used for distribution) in domain $d$ |
| $A, h_A$ | Subgroup $A \subseteq \mathcal{X}$ and binary indicator function for subgroup $A$ |
| $\ell(y, f(x))$ | Loss per data point $(x, y)$ |
| $Z_d(x), d \in \{0,1\}$ | Average loss per data point $E_d[\ell(Y, f(X))|X = x]$ |
| $E_d[\cdot|X \in A]$ | Expectation with respect to $p_d$ for a subgroup |
| $\tau$ | Tolerance specified by analyst to detect performance decay of magnitude $\tau$ or higher |
| $\epsilon$ | Prevalence specified by analyst for the minimum group size to detect |
| $\alpha$ | Significance level or the desired false rejection rate of the null hypothesis |
| $\mu_d(x)$ | conditional outcome function in source domain, $\mu_d(x) = p_d(Y = 1|X = x)$ |
| $p_s(y|x_s, \mu_0), Z_s(x)$ | Conditional outcome distribution shifted with respect to $X_s$ and the resulting loss |
| $Z_{0,s}(x)$ | Conditional loss function $Z_{0,s}(x) = E_0[\ell|x_s, h_A(x)]$ |
| $\pi(x), \pi_s(x)$ | Density ratios $\frac{p_1(x)}{p_0(x)}$ and $\pi_s(x) = \frac{p_1(x_s)}{p_0(x_s)}$ |
| $\pi_A(x), \pi_{s,A}(x)$ | Scaled versions of $\pi(x)$ and $\pi_s(x)$ by the factors $\frac{E_0[h_A(X)]}{E_1[h_A(X)]}$ and $\frac{E_0[h_A(X)]}{E_s[h_A(X)]}$ respectively |

## A. Connection to causal mediation analysis

Figure 4 summarizes the correspondence between the proposed tests and the tests for interaction and mediation in the causal inference literature. Proposed hypothesis tests could benefit heterogeneous mediation analysis that identifies meaningful subgroups, as opposed to prespecified or model-driven latent subgroups. Specifically, the proposed tests can be extended to determine whether a subgroup with large direct and indirect effects exists. The extension is straightforward when subgroups are defined using pre-exposure covariates, as in conventional mediation analysis. When subgroups are defined using potential mediators, further identification assumptions are required before the proposed tests can be applied.

## B. Modified covariate shift test

Counter-intuitively, the tests as defined in Hypotheses 3.1 and 3.3 may flag shifts in covariates that are unrelated to the loss function. That is, for the aggregate covariate shift test, we may find a group $A$ for which

$$E_1[Z_0(X)|X \in A] - E_0[Z_0(X)|X \in A] > 0$$

even when $Z_0(X) = Z_1(X)$. This can happen because we are conditioning on $X \in A$ which introduces a shift in $\ell|X \in A$ by conditioning on the collider $A$ for effect of $d$ on $\ell$. A simple fix is to restrict the detectors to depend only on features that

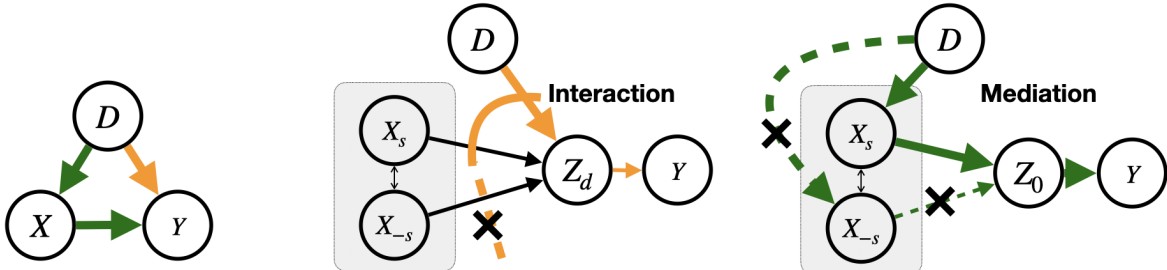

Figure 4: **Graphical description of the hypothesis tests.** Differences in distributions of $X$ and $Y$ (and hence average performance of a model) across the two domains $D = 0, 1$ is represented by effect of variable $D$ on $X, Y$. (**left**) Performance can vary due to changes in conditional outcome distribution (orange edge, outcome shift) and/or changes in the covariate distribution (green path, covariate shift). Variables $Z_d, d \in \{0, 1\}$ represents conditional loss in the domains as described in text. Firstly, we test whether each of the effects is zero. (**middle**) When test for outcome shift is rejected, we identify feature subsets $X_s$ such that the complement $X_{-s}$ do not interact with the effect of $D$ on $Y$ (orange dashed line). (**right**) When test for covariate shift is rejected, we identify feature subsets $X_s$ such that $X_{-s}$ which do not mediate the effect of $D$ on $Y$ (green dashed path).

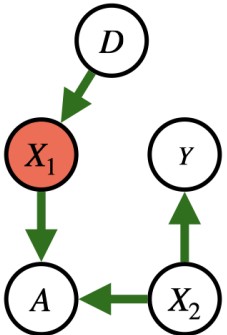

Figure 5: **Modified covariate shift test.** Graph provides an example data generating mechanism where $X_1$ is marginally independent of $Y$. The feature $X_2$ influences $Y$ but does not shift. Conditioning on $A$ introduces a collider on the path from $D$ to $Y$ (and hence $\ell$). Therefore, the loss is affected by the covariate shift even though it only depends on a feature that does not shift. In the modified test, we remove the feature $X_1$ before testing for covariate shifts.

are correlated with the loss.

For the covariate shift tests, we subset the covariates to keep those that are correlated with the loss, $X_{corr} = \{X_i | X_i \not\perp \ell, i \in \{1, 2, \ldots, |X|\}\}$. The null hypotheses 3.1 and 3.3 are defined only with respect to the correlated features, $X := X_{corr}$. To avoid 'peeking into the data', we filter uncorrelated features based on the training data split. Remaining part of the tests are the same as earlier.

## C. Step-by-step testing procedures

Source and target domains comprise of i.i.d. observations of $(x, y)$. We denote empirical average over $n$ data points from domain $d$ by the operator $\mathbb{P}_{d,n}$. For example, $\mathbb{P}_{0,n}$ refers to taking the empirical average for the evaluation dataset of the source domain. We use $\mathbb{P}_n$ to refer to an average over the pooled source and target data. For ease in notation, we write the same number of samples $n$ for each dataset. However, they could be different. In fact, our hypothesis tests are particularly beneficial for cases when the target datasets are too small for point-identifying performance shifts. We split the datasets into two parts, one is used for fitting all nuisance parameters and the detectors, and other is used to evaluate the MEE and for inference.

### C.1. Fitting nuisance parameters

We require seven nuisance parameters denoted by $\eta = (Z_0, Z_1, Z_s, Z_{0,s}, \pi, \pi_s, \pi_V)$. We denote their estimates by $\hat{\eta}$. The nuisance parameters can be broadly categorized as outcome and density ratio models.

- *Outcome models.* To fit the conditional loss functions $\hat{Z}_0$ and $\hat{Z}_1$, we first fit the conditional outcome probabilities $\hat{p}_0(y|x)$ and $\hat{p}_1(y|x)$ by regressing $y$ on $x$. Then, conditional loss is simply an expectation over the two possible outcome $\hat{Z}_d(x) = \sum_{y \in \{0,1\}} \ell(y, f(x)) \hat{p}_d(y|x)$. For $\hat{Z}_s$, we fit the $X_s$-shifted conditional outcome probability $\hat{p}_s$ by regressing $y$ on $x_s, \hat{\mu}_0(x)$, and taking an expectation over $y$. That is, $\hat{Z}_s(x) = \sum_{y \in \{0,1\}} \ell(y, f(x)) \hat{p}_s(y|x_s, \hat{\mu}_0(x))$. Similarly, the conditional loss function $Z_{0,s}(x) = E_0[\ell | x_s, h_A(x)]$ can be estimated through regressing $y$ on $x_s, h_A(x)$.
- *Density ratio models.* We estimate the scaled density ratios $\tilde{\pi}_A(x) = \frac{p_1(x) E_0[h_A(X)]}{p_0(x) E_1[h_A(X)]}$ and $\tilde{\pi}_{s,A}(x) = \frac{p_1(x_s) E_0[h_A(X)]}{p_0(x_s) E_s[h_A(X)]}$ by first fitting models for the density ratios and plugging them into the definitions. We also need a density ratio model for the detailed outcome test, $\pi_V(X_s = x_s, X_{-s} = x_{-s}) = \frac{p_1(X_{-s} = x_{-s} | X_s = x_s, \mu_0(X) = \mu_0(x))}{p_1(X_{-s} = x_{-s} | \mu_0(X) = \mu_0(x))}$. Density ratios are fit using a probabilistic classifier to estimate the probability a data point belongs to target domain from the pooled source and target data. Density ratios are computed as odds ratio for the classifiers. Please refer to Sugiyama et al. (2007) for more details.

*Remark.* We note that the scaled density ratio models depend on the detectors $h$ which we haven't fit yet. To break this cyclical dependence, we first fit detectors (as shown in the next section) for the *unscaled* density ratios and then compute the scaled density ratio models.

### C.2. Fitting detectors

Recall that detectors $h_A(x)$ are binary functions meant to detect the data points with high mean exceedence. Fitting procedures for detectors vary for the tests since the exceedence is defined differently. All detectors can be implemented using off-the-shelf ML regression libraries.

**Aggregate test.** For the outcome test, we use the plug-in approach $h_A(x) = \mathbb{1}\{\hat{Z}_1(x) - \hat{Z}_0(x) - \tau > 0\}$. Alternatively, we can fit a detector by regressing $\ell - \hat{Z}_0(x) - \tau$ from $x$,

$$h_A \in \arg \min_g \mathbb{P}_{1,n}((\ell - \hat{Z}_0(x) - \tau) - g(x))^2 \tag{10}$$

For the covariate test, we solve the optimization problem (9) after plugging in $\hat{Z}_0$ and $\hat{\pi}$. We find values of $\omega, \lambda$ by grid search that maximize the objective for detectors of the form,

$$h_A^{(\omega, \lambda)} = \left\{ x : \left( \hat{Z}_0(x) - \lambda \right) (\hat{\pi}(x) \omega - 1) \geq \tau \right\}. \tag{11}$$

**Detailed test.** For outcome test, we simply plug-in the estimated nuisance parameters into the MEE and threshold it,

$$h_A(x) = \mathbb{1}\{\hat{Z}_1(x) - \hat{Z}_s(x) - \tau > 0\}. \tag{12}$$

For the covariate test, we grid search for $\omega, \lambda$ values to optimize the detailed test counterpart of (9) with detectors of the form,

$$h_{s,A}^{(\omega,\lambda)} = \left\{ x : \left( \hat{Z}_0(x) - \lambda \right) (\hat{\pi}_A(x)\omega - \hat{\pi}_{s,A}(x)) \geq \tau\hat{\pi}_{s,A}(x) \right\}. \tag{13}$$

### C.3. Estimating the Maximum conditional Expectation of the Exceedence (McEE)

The main manuscript discusses the testing procedure in terms of the MEE for ease of exposition. In practice, to maximize statistical power for the hypothesis tests in SHIFT, it is better to construct estimators of the Maximum conditional Expectation of the Exceedence (McEE), as its efficient estimator has lower variance. Below, we present the inference procedure for McEE on the held-out data, which is slightly more complex than that for MEE because it involves ratios instead.

We first present plug-in estimators for McEE, which are key quantities in the tests (Table 5). However, as noted earlier, plug-in estimates have been shown to give biased estimates for estimands involving infinite-dimensional quantities like outcome models or density ratios, and using flexible ML estimators for the nuisance parameters do not help remove the bias (Chernozhukov et al., 2018). For this reason, the plug-in estimators cannot be readily used to perform valid tests. We propose to debias the estimates by *one-step correction*, also known as double/debiased estimation. Debiased estimators can be shown to be asymptotically normal and give valid tests. Note that the estimators for aggregate are special cases of the ones for detailed tests with $X_s$ set to an empty set. Therefore, we only present the estimators for the detailed tests and then analyze their theoretical properties.

Table 5: Plug-in and debiased estimates of restricted MEE for detector $h$ given the fitted nuisance parameters $\hat{\eta}$. We recommend using the debiased estimates because they can give valid tests.

| Type of test | Plug-in estimator | Debiased estimator |
|---|---|---|
| Aggregate outcome | | same as $\text{McEE}_Y(\emptyset)$ |
| | $\mathbb{P}_{1,n}[(\ell(Y, f(X)) - \hat{Z}_0(X) - \tau)h_A(X)]/\mathbb{P}_{1,n}[h(X)]$ | |
| Aggregate covariate | | same as $\text{McEE}_X(\emptyset)$ |
| | $\mathbb{P}_{1,n}\hat{Z}_0(x)h_A(x)/\mathbb{P}_{1,n}h_A(x) - \mathbb{P}_{0,n}[\ell h_A(x)]/\mathbb{P}_{0,n}[h_A(x)] - \tau$ | |
| Detailed outcome, $\text{McEE}_Y(s)$ | | given in (14) |
| | $\mathbb{P}_{1,n}[(\ell(Y, f(X)) - \hat{Z}_s(X) - \tau)h_A(X)]/\mathbb{P}_{1,n}[h_A(X)]$ | |
| Detailed covariate, $\text{McEE}_X(s)$ | | given in (17) |
| | $\mathbb{P}_{1,n}\hat{Z}_0(x)h_A(x)/\mathbb{P}_{1,n}h_A(x) - \mathbb{P}_{0,n}[\hat{\pi}_{s,A}(x)\ell h_A(x)]/\mathbb{P}_{0,n}[h_A(x)] - \tau$ | |

**Detailed test for outcome shift.** As discussed in Sec 4, the MEE in the case of detailed outcome shift is not pathwise differentiable because it depends on an indicator function (in $Z_s(x)$). Therefore, we follow the binning trick of Singh et al. (2024) to change MEE into a pathwise differentiable quantity. Recall that $X_s$-specific outcome shifts are defined as a function of $\mu_0(x)$. We change it to depend on a binned version, $\mu_{\text{bin}}(x) = \frac{1}{B}\lfloor\mu_0(x)B + \frac{1}{2}\rfloor$ for some fixed $B \in \mathbb{Z}_+$. It discretized $\mu_0(x)$ into $B$ equally spaced bins in $[0,1]$. Thus, the binned MEE is defined as a function of $Z_{s,\text{bin}} = \sum_y \ell(y, f(x))p_s(y|x_s, \mu_{\text{bin}}(x))$. As long as the binned $\mu_{\text{bin}}$ does not fall on the bin edges (almost surely), the indicator involved in the binned MEE is zero (almost surely). Thus, ensuring that MEE is pathwise differentiable. We expect the binned and original version of MEE to be similar for a large enough number of bins. We can now define the estimator.

The debiased estimator for MEE has the form of a V-statistic. We denote V-statistic by the operator $\mathbb{P}_{1,n}\tilde{\mathbb{P}}_{1,n}$, which takes an average over all pairs of observations with replacement. That is, V-statistic over observations $O_1, O_2, \ldots, O_n$ is

computed as $\frac{1}{n^2}\sum_{i=1}^{n}\sum_{j=1}^{n}v(O_i, O_j)$ for some function $v$. The tilde notation $\tilde{\mathbb{P}}_{1,n}$ is used to distinguish between the arguments to $v$. Debiased estimator for MEE given the fitted nuisance parameters $\hat{\eta}$ is expressed as a ratio $\widehat{\mathrm{McEE}}_{\mathrm{Y}}(s) = \widehat{\mathrm{McEE}}_{\mathrm{Y}}^{\mathrm{num}}(s)/\mathbb{P}_n[h_A(x)]$ where $\widehat{\mathrm{McEE}}_{\mathrm{Y}}^{\mathrm{num}}(s)$ is

$$\widehat{\mathrm{McEE}}_{\mathrm{Y}}^{\mathrm{num}}(s) = \mathbb{P}_{1,n}\tilde{\mathbb{P}}_{1,n}\ell(Y, f(X_s, \tilde{X}_{-s}))h_A(X_s, \tilde{X}_{-s})\hat{\pi}_V(X_s, \tilde{X}_{-s}|\hat{\mu}_{\mathrm{bin}}) \tag{14}$$

$$+ \mathbb{P}_{1,n}\sum_y \ell(y, f(X))h_A(X)\hat{p}_1(y|X_s, \hat{\mu}_{\mathrm{bin}}(X)) \tag{15}$$

$$- \mathbb{P}_{1,n}\sum_y \ell(y, f(X_s, \tilde{X}_{-s})h_A(X_s, \tilde{X}_{-s})\hat{\pi}_V(X_s, \tilde{X}_{-s}|\hat{\mu}_{\mathrm{bin}}(X))\hat{p}_1(y|X_s, \hat{\mu}_{\mathrm{bin}}(X)) \tag{16}$$

**Detailed test for covariate shift.** Define the conditional loss function $Z_{0,s}(x) = E_0[\ell|x_s, h_A(x)]$ and its estimate as $\hat{Z}_{0,s}$. Debiased estimator for MEE in the case of covariate shift given the fitted nuisance parameters $\hat{\eta}$ is expressed as a ratio $\widehat{\mathrm{McEE}}_{\mathrm{X}}(s) = \widehat{\mathrm{McEE}}_{\mathrm{X}}^{\mathrm{num}}(s)/\mathbb{P}_n[h_A(x)]$ where $\widehat{\mathrm{McEE}}_{\mathrm{X}}^{\mathrm{num}}(s)$ is

$$\widehat{\mathrm{McEE}}_{\mathrm{X}}^{\mathrm{num}}(s) = \mathbb{P}_{1,n}[\hat{Z}_0(x)h_A(x)] - \mathbb{P}_{0,n}[\ell\hat{\pi}_{s,A}(x)] \tag{17}$$

$$+ \mathbb{P}_{0,n}[(\ell - \hat{Z}_0(x))\hat{\pi}_A(x)] \tag{18}$$

$$+ \mathbb{P}_{0,n}[\hat{Z}_{0,s}(x)\hat{\pi}_{s,A}(x)] - \mathbb{P}_{1,n}[\hat{Z}_{0,s}(x)] \tag{19}$$

### C.4. Inference

For inference, we use the Gaussian multiplier bootstrap method to construct bootstrap distributions for the test statistic (McEE) under the null (Hsu, 2017). Each bootstrap sample, involves sampling $n$ variables distributed as standard normal, $\xi_1, \ldots, \xi_n \sim N(0, 1)$ and recomputing the centered McEE with $\xi$ as per-sample weights. We construct the $p$-value as the proportion of times the bootstrap test statistics exceeds the observed test statistic.

## D. Theoretical results

Here, we provide details of type-I error and power guarantees of our testing procedures for the four hypotheses, two aggregate (covariate/outcome) and two detailed level (covariate/outcome for a subset). Specifically, we analyze the McEE for the restricted version of the tests with a singleton detector $\{h_A\}$. Extension to maxima over multiple detectors follows from Gaussian approximations of maxima over averages (e.g. Chernozhukov et al. (2013)).

**Outline.** At a high-level, we show that our debiased estimators of McEE are asymptotically linear. It implies that they converge to a normal distribution centered at the true McEE. Because of the normality, we can conduct inference using standard tests such as z-test or Wald test for the null hypothesis McEE $\leq 0$. Properties of type-I error control and power will follow. So, the key is to show asymptotic linearity of the estimators.

We first give an outline of the linearity analysis, which applies the techniques developed for analyzing V-statistics (van der Vaart, 1998). Our estimators of McEE, (14) and (17), are one-step corrected estimators which start from a plug-in estimate, McEE($\hat{\mathbb{P}}$), and debias it by adding its canonical gradient $\mathbb{P}_n\psi(o; \hat{\mathbb{P}})$ (also called an influence function).

$$\mathrm{McEE}(\hat{\mathbb{P}}) + \mathbb{P}_n\psi(o; \hat{\mathbb{P}})$$

Here, the canonical gradient $\psi$ is a function of the probability distribution $\mathbb{P}$ over the observations $O = (X, Y, D)$. Following the von-Mises expansion of McEE($\hat{\mathbb{P}}$), the bias of the one-step corrected estimator can be decomposed into three terms,

$$\left(\mathrm{McEE}(\hat{\mathbb{P}}) + \mathbb{P}_n\psi(o; \hat{\mathbb{P}})\right) - \mathrm{McEE}(\mathbb{P})$$

$$= (\mathbb{P}_n - \mathbb{P})\psi(o; \hat{\mathbb{P}}) + R(\hat{\mathbb{P}}, \mathbb{P})$$

$$= (\mathbb{P}_n - \mathbb{P})\psi(o; \mathbb{P}) + (\mathbb{P}_n - \mathbb{P})(\psi(o; \hat{\mathbb{P}}) - \psi(o; \mathbb{P})) + R(\hat{\mathbb{P}}, \mathbb{P})$$

for a second-order remainder term $R(\hat{\mathbb{P}}, \mathbb{P})$. The goal of our analysis is to show that each of the terms is negligible at $o_p(n^{-1/2})$ rate such that we get an asymptotically linear representation of the one-step corrected estimator.

$$\left(\mathrm{McEE}(\hat{\mathbb{P}}) + \mathbb{P}_n\psi(o; \hat{\mathbb{P}})\right) - \mathrm{McEE}(\mathbb{P}) = \mathbb{P}_n\psi(o; \mathbb{P}) + o_p(n^{-1/2}).$$

The form implies that the one-step corrected estimator is asymptotically normal with mean $\text{McEE}(\mathbb{P})$ and variance $var(\psi(o;\mathbb{P}))/n$, which allows constructing valid tests such as z-test or Wald test.

We present the theoretical results for detailed tests in the next section.

### D.1. Detailed test of outcome shift

For pathwise differentiability of the McEE, we require a mild condition to hold for the binned outcome shift $\mu_{\text{bin}}$. It states that the true shifted probabilities $\mu_{\text{bin}}$ lie *inside* the bins almost surely. Additionally, we require that the nuisance parameters are consistently estimated. It is possible to ensure consistency by using nonparametric ML estimators. Most importantly, we require that the product of estimation errors in the V-statistic density ratio model $\hat{\pi}_V$ and the $X_s$-specific outcome shift model converges at $o_p(n^{-1/2})$ rate. In the average treatment effect literature, similar assumptions is made for the outcome and treatment propensity models (Chernozhukov et al., 2018). Note that this condition is significantly milder than requiring that both the models have fast convergence. The condition holds as long as we can guarantee $o_p(n^{-1/4})$ rate of convergence for the models.

**Condition D.1.** For variable subset $s$, assume the following holds:

- (binning) For all bin edges $b$ of $\mu_0(x)$ except $\{0, 1\}$, the set $\{x : |\mu_0(x) - b| \leq \epsilon\}$ is measure zero for some $\epsilon$.
- (consistency) Nuisance parameter estimates $\hat{\mu}_{\text{bin}}, \hat{\pi}_V, \hat{p}_s$ are consistent.
- (product of errors) $\mathbb{P}_1(\hat{\pi}_V - \pi_V)(\hat{p}_s - p_s) = o_p(n^{-1/2})$.

**Theorem D.2.** *Suppose Condition D.1 holds. Then the estimator $\widehat{\text{McEE}}_Y(s)$ follows a normal distribution asymptotically, centered at the estimand $McEE_Y(s)$.*

To prove the above theorem, recall that $\text{McEE}_Y(s)$ is defined as the ratio, $\text{McEE}_Y^{\text{num}}(s)/\mathbb{P}_n[h_A(x)]$. The following lemma will first show that the estimator for the numerator is asymptotically linear. Denominator can be estimated simply as an empirical average of $h_A(x)$, hence it is linear. Then, we will use the Delta Method (van der Vaart, 1998) to estimate the ratio and prove it has a normal distribution asymptotically.

Suppose the nuisance parameters in $\widehat{\text{McEE}}_Y^{\text{num}}(s)$ are $\eta_{Y,s}^{\text{num}} = (\pi_V, \mu_{\text{bin}}, p_s)$. We write the one-step corrected estimate of $\widehat{\text{McEE}}_Y^{\text{num}}(s)$ as a V-statistic.

$$
\begin{aligned}
\widehat{\text{McEE}}_Y^{\text{num}}(s) = {}& \mathbb{P}_{1,n}\tilde{\mathbb{P}}_{1,n}\ell(Y, f(X_s, \tilde{X}_{-s}))h_A(X_s, \tilde{X}_{-s})\hat{\pi}_V(X_s, \tilde{X}_{-s}|\hat{\mu}_{\text{bin}}) \\
& + \mathbb{P}_{1,n}\tilde{\mathbb{P}}_{1,n}\sum_y \ell(y, f(X))h_A(X)\hat{p}_1(y|X_s, \hat{\mu}_{\text{bin}}(X)) \\
& - \mathbb{P}_{1,n}\tilde{\mathbb{P}}_{1,n}\sum_y \ell(y, f(X_s, \tilde{X}_{-s})h_A(X_s, \tilde{X}_{-s})\hat{\pi}_V(X_s, \tilde{X}_{-s}|\hat{\mu}_{\text{bin}}(X))\hat{p}_1(y|X_s, \hat{\mu}_{\text{bin}}(X)) \\
=: {}& \mathbb{P}_{1,n}\tilde{\mathbb{P}}_{1,n}v(X, Y, \tilde{X}, \tilde{Y}; \hat{\eta}_{Y,s}^{\text{num}})
\end{aligned} \tag{20}
$$

**Lemma D.3.** *Assuming Condition D.1 holds, $\widehat{\text{McEE}}_Y^{\text{num}}(s)$ is an asymptotically linear estimator for $McEE_Y^{\text{num}}(s)$*

$$
\widehat{\text{McEE}}_Y^{\text{num}}(s) - McEE_Y^{\text{num}}(s) = \mathbb{P}_{1,n}\psi(x, y; \hat{\eta}_{Y,s}^{\text{num}}) + o_p(n^{-1/2})
$$

*for the influence function*

$$
\psi(x, y; \hat{\eta}_{Y,s}^{\text{num}}) = \mathbb{P}_1\ell(y, f(x_s, X_{-s}))h_A(x_s, X_{-s})\pi_V(x_s, X_{-s}|\mu_{bin}) \tag{21}
$$

$$
+ \sum_{\tilde{y}} \ell(\tilde{y}, f(x))h_A(x)p_1(\tilde{y}|x_s, \mu_{bin}(x)) \tag{22}
$$

$$
- \sum_{\tilde{y}} \ell(\tilde{y}, f(x_s, \tilde{x}_{-s})h_A(x_s, \tilde{x}_{-s})\pi_V(X_s, \tilde{X}_{-s}|\mu_{bin}(X))p_1(\tilde{y}|x_s, \mu_{bin}(x)) \tag{23}
$$

$$
\tag{24}
$$

*Proof.* Define the symmetrized version of $v$ in (20) as $v_{sym}(X, Y, \tilde{X}, \tilde{Y}) = \frac{v(X,Y,\tilde{X},\tilde{Y}) + v(\tilde{X},\tilde{Y},X,Y)}{2}$. Then, we rewrite the estimator as

$$
\widehat{\text{McEE}}_Y^{\text{num}}(s) = \mathbb{P}_{1,n}\tilde{\mathbb{P}}_{1,n}v_{sym}\left(X, Y, \tilde{X}, \tilde{Y}; \hat{\eta}_{Y,s}^{\text{num}}\right).
$$

Following Theorem 12.3 in (van der Vaart, 1998), the Hájek projection of $\widehat{\mathrm{McEE}}_{\mathrm{Y}}^{\mathrm{num}}(s)$ is

$$
\begin{aligned}
\hat{u}_{\mathrm{Y}}^{\mathrm{num}}(s) &= \sum_{i=1}^{n} \mathbb{P}_1 \left[ \mathbb{P}_{1,n}\tilde{\mathbb{P}}_{1,n} v_{sym}\left(X, Y, \tilde{X}, \tilde{Y}; \hat{\eta}_{\mathrm{Y},s}^{\mathrm{num}}\right) - \widehat{\overline{\mathrm{McEE}}}_{Y}^{\mathrm{num}}(s) \mid X_i, Y_i \right] \\
&= \sum_{i=1}^{n} \mathbb{P}_1 \left[ v_{sym}\left(X_i, Y_i, X^{(2)}, Y^{(2)}; \hat{\eta}_{\mathrm{Y},s}^{\mathrm{num}}\right) - \widehat{\overline{\mathrm{McEE}}}_{Y}^{\mathrm{num}}(s) \mid X_i, Y_i \right] \\
&= \sum_{i=1}^{n} v_{sym,1}\left(X_i, Y_i; \hat{\eta}_{\mathrm{Y},s}^{\mathrm{num}}\right)
\end{aligned}
$$

where $\widehat{\overline{\mathrm{McEE}}}_{Y}^{\mathrm{num}}(s) = \mathbb{P}_1\tilde{\mathbb{P}}_1 v_{sym}\left(X, Y, \tilde{X}, \tilde{Y}; \hat{\eta}_{\mathrm{Y},s}^{\mathrm{num}}\right)$.

Decompose the bias of the estimator as,

$$
\begin{aligned}
\widehat{\mathrm{McEE}}_{\mathrm{Y}}^{\mathrm{num}}(s) - \mathrm{McEE}_{\mathrm{Y}}^{\mathrm{num}}(s) =& \mathbb{P}_{1,n}\tilde{\mathbb{P}}_{1,n} v_{sym}\left(X, Y, \tilde{X}, \tilde{Y}; \hat{\eta}_{\mathrm{Y},s}^{\mathrm{num}}\right) - \mathbb{P}_1\tilde{\mathbb{P}}_1 v_{sym}\left(X, Y, \tilde{X}, \tilde{Y}; \eta_{\mathrm{Y},s}^{\mathrm{num}}\right) \\
=& \mathbb{P}_{1,n}\tilde{\mathbb{P}}_{1,n} v_{sym}\left(X, Y, \tilde{X}, \tilde{Y}; \hat{\eta}_{\mathrm{Y},s}^{\mathrm{num}}\right) - \mathbb{P}_{1,n}\left[ h_{sym,1}\left(X, Y; \hat{\eta}_{\mathrm{Y},s}^{\mathrm{num}}\right) + \widehat{\overline{\mathrm{McEE}}}_{Y}^{\mathrm{num}}(s) \right] & (25) \\
&+ (\mathbb{P}_{1,n} - \mathbb{P}_1)\left( v_{sym,1}\left(X, Y; \hat{\eta}_{\mathrm{Y},s}^{\mathrm{num}}\right) + \widehat{\overline{\mathrm{McEE}}}_{Y}^{\mathrm{num}}(s) - v_{sym,1}(X, Y) - \mathrm{McEE}_{\mathrm{Y}}^{\mathrm{num}}(s) \right) \\
& & (26) \\
&+ (\mathbb{P}_{1,n} - \mathbb{P}_1)\left( v_{sym,1}(X, Y) + \mathrm{McEE}_{\mathrm{Y}}^{\mathrm{num}}(s) \right) & (27) \\
&+ \mathbb{P}_1\left( v_{sym,1}\left(X, Y; \hat{\eta}_{\mathrm{Y},s}^{\mathrm{num}}\right) + \widehat{\overline{\mathrm{McEE}}}_{Y}^{\mathrm{num}}(s) - v_{sym,1}\left(X, Y; \eta_{\mathrm{Y},s}^{\mathrm{num}}\right) - \mathrm{McEE}_{\mathrm{Y}}^{\mathrm{num}}(s) \right). \\
& & (28)
\end{aligned}
$$

Now, we consider the four terms of the decomposition one-by-one.

Term (25): Suppose $\mathbb{P}_1 v_{sym}^2(X, Y, \tilde{X}, \tilde{Y}; \hat{\eta}_{\mathrm{Y},s}^{\mathrm{num}}) < \infty$. Via a straightforward extension of the proof in Theorem 12.3 in van der Vaart (1998), one can show that

$$
\frac{var\left( \mathbb{P}_{1,n}\tilde{\mathbb{P}}_{1,n} v_{sym}\left(X, Y, \tilde{X}, \tilde{Y}; \hat{\eta}_{\mathrm{Y},s}^{\mathrm{num}}\right)\right)}{var\left( \mathbb{P}_{1,n} v_{sym,1}\left(X, Y; \hat{\eta}_{\mathrm{Y},s}^{\mathrm{num}}\right)\right)} \to_p 1.
$$

Then by Theorem 11.2 in van der Vaart (1998) and Slutsky's lemma, we have

$$
\mathbb{P}_{1,n}\tilde{\mathbb{P}}_{1,n} v_{sym}\left(X, Y, \tilde{X}, \tilde{Y}; \hat{\eta}_{\mathrm{Y},s}^{\mathrm{num}}\right) - \mathbb{P}_{1,n}\left[ v_{sym,1}\left(X, Y; \hat{\eta}_{\mathrm{Y},s}^{\mathrm{num}}\right) + \widehat{\overline{\mathrm{McEE}}}_{Y}^{\mathrm{num}}(s) \right] = o_p\left(n^{-1/2}\right).
$$

Term (26): This term is asymptotically negligible since we perform sample splitting to estimate the nuisance parameters and evaluate the estimator for $\widehat{\overline{\mathrm{McEE}}}_{Y}^{\mathrm{num}}(s)$ on separate datasets. Then by Lemma 1 in Kennedy (2024), we have that

$$
(\mathbb{P}_{1,n} - \mathbb{P}_1)\left( v_{sym,1}\left(X, Y; \hat{\eta}_{\mathrm{Y},s}^{\mathrm{num}}\right) + \widehat{\overline{\mathrm{McEE}}}_{Y}^{\mathrm{num}}(s) - v_{sym,1}\left(X, Y; \eta_{\mathrm{Y},s}^{\mathrm{num}}\right) - \mathrm{McEE}_{\mathrm{Y}}^{\mathrm{num}}(s) \right) = o_p(n^{-1/2})
$$

as long as the estimators for the nuisance parameters are consistent.

Term (27): This term is the difference between empirical and population average of a population-level quantity, hence it follows a normal distribution asymptotically by standard CLT.

Term (28): We show that the term is asymptotically negligible as well given the condition on nuisance estimation errors.

$$\mathbb{P}_1\tilde{\mathbb{P}}_1\left(v_{sym}\left(X,Y,\tilde{X},\tilde{Y};\hat{\eta}^{\mathrm{num}}_{Y,s}\right)+\widehat{\overline{\mathrm{McEE}}}_Y^{\mathrm{num}}(s)-v_{sym}\left(X,Y,\tilde{X},\tilde{Y};\eta^{\mathrm{num}}_{Y,s}\right)-\mathrm{McEE}^{\mathrm{num}}_Y(s)\right)$$

$$=\mathbb{P}_1\tilde{\mathbb{P}}_1\left(v(X,Y,\tilde{X},\tilde{Y};\hat{\eta}^{\mathrm{num}}_{Y,s})-v(X,Y,\tilde{X},\tilde{Y};\eta^{\mathrm{num}}_{Y,s})\right) \tag{29}$$

$$=\mathbb{P}_1\tilde{\mathbb{P}}_1\ell(Y,f(X_s,\tilde{X}_{-s}))h_A(X_s,\tilde{X}_{-s})\left(\hat{\pi}_V(X_s,\tilde{X}_{-s}|\hat{\mu}_{\mathrm{bin}})-\pi_V(X_s,\tilde{X}_{-s}|\mu_{\mathrm{bin}})\right)$$

$$+\mathbb{P}_1\sum_y\ell(y,f(X))h_A(X)\left(\hat{p}_1(y|X_s,\hat{\mu}_{\mathrm{bin}}(X))-p_1(y|X_s,\mu_{\mathrm{bin}}(X))\right)$$

$$-\mathbb{P}_1\tilde{\mathbb{P}}_1\sum_y\ell(y,f(X_s,\tilde{X}_{-s})h_A(X_s,\tilde{X}_{-s}) \tag{30}$$

$$\times\left(\hat{\pi}_V(X_s,\tilde{X}_{-s}|\hat{\mu}_{\mathrm{bin}}(X))\hat{p}_1(y|X_s,\hat{\mu}_{\mathrm{bin}}(X))-\pi_V(X_s,\tilde{X}_{-s}|\mu_{\mathrm{bin}}(X))p_1(y|X_s,\mu_{\mathrm{bin}}(X))\right) \tag{31}$$

By the law of iterated expectation, we can expand the first summand as,

$$=\mathbb{P}_1\tilde{\mathbb{P}}_1\ell(Y,f(X_s,\tilde{X}_{-s}))h_A(X_s,\tilde{X}_{-s})\left(\hat{\pi}_V(X_s,\tilde{X}_{-s}|\hat{\mu}_{\mathrm{bin}})p_1(y|X_s,\mu_{\mathrm{bin}}(X))-\pi_V(X_s,\tilde{X}_{-s}|\mu_{\mathrm{bin}})p_1(y|X_s,\mu_{\mathrm{bin}}(X))\right)$$

$$+\mathbb{P}_1\sum_y\ell(y,f(X))h_A(X)\left(\hat{p}_1(y|X_s,\hat{\mu}_{\mathrm{bin}}(X))-p_1(y|X_s,\mu_{\mathrm{bin}}(X))\right)$$

$$-\mathbb{P}_1\tilde{\mathbb{P}}_{1,n}\sum_y\ell(y,f(X_s,\tilde{X}_{-s})h_A(X_s,\tilde{X}_{-s}) \tag{32}$$

$$\times\left(\hat{\pi}_V(X_s,\tilde{X}_{-s}|\hat{\mu}_{\mathrm{bin}}(X))\hat{p}_1(y|X_s,\hat{\mu}_{\mathrm{bin}}(X))-\pi_V(X_s,\tilde{X}_{-s}|\mu_{\mathrm{bin}}(X))p_1(y|X_s,\mu_{\mathrm{bin}}(X))\right)$$

Cancelling terms from the first and third estimand gives us,

$$=-\mathbb{P}_1\tilde{\mathbb{P}}_1\ell(Y,f(X_s,\tilde{X}_{-s}))h_A(X_s,\tilde{X}_{-s})\hat{\pi}_V(X_s,\tilde{X}_{-s}|\hat{\mu}_{\mathrm{bin}})\left(\hat{p}_1(y|X_s,\hat{\mu}_{\mathrm{bin}}(X))-p_1(y|X_s,\mu_{\mathrm{bin}}(X))\right)$$

$$+\mathbb{P}_1\sum_y\ell(y,f(X))h_A(X)\left(\hat{p}_1(y|X_s,\hat{\mu}_{\mathrm{bin}}(X))-p_1(y|X_s,\mu_{\mathrm{bin}}(X))\right)$$

By the definition of the density ratio $\pi_V$, the second summand can written as a V-statistic.

$$=-\mathbb{P}_1\tilde{\mathbb{P}}_1\ell(Y,f(X_s,\tilde{X}_{-s}))h_A(X_s,\tilde{X}_{-s})\hat{\pi}_V(X_s,\tilde{X}_{-s}|\hat{\mu}_{\mathrm{bin}})\left(\hat{p}_1(y|X_s,\hat{\mu}_{\mathrm{bin}}(X))-p_1(y|X_s,\mu_{\mathrm{bin}}(X))\right)$$

$$+\mathbb{P}_1\tilde{\mathbb{P}}_1\sum_y\ell(y,f(X))h_A(X)\pi_V(X_s,\tilde{X}_{-s}|\mu_{\mathrm{bin}}(X))\left(\hat{p}_1(y|X_s,\hat{\mu}_{\mathrm{bin}}(X))-p_1(y|X_s,\mu_{\mathrm{bin}}(X))\right)$$

Collecting the common terms, the sum simplifies to

$$=\mathbb{P}_1\tilde{\mathbb{P}}_1\sum_y\ell(y,f(X))h_A(X) \tag{33}$$

$$\times\left(\pi_V(X_s,\tilde{X}_{-s}|\mu_{\mathrm{bin}}(X))-\hat{\pi}_V(X_s,\tilde{X}_{-s}|\hat{\mu}_{\mathrm{bin}}(X))\right)\left(\hat{p}_1(y|X_s,\hat{\mu}_{\mathrm{bin}}(X))-p_1(y|X_s,\mu_{\mathrm{bin}}(X))\right),$$

which is $o_p(n^{-1/2})$ by Condition (D.1) on product of errors in $\hat{\pi}_V$ and $\hat{p}_s$ models.

Hence, the bias of the estimator is $o_p(n^{-1/2})$. $\qquad\square$

*Proof of Theorem D.2.* Applying Delta Method to the numerator and denominator estimates, which we know to be asymptotically linear from Lemma D.3, we get that the estimator $\widehat{\mathrm{McEE}}_Y(s)=\widehat{\overline{\mathrm{McEE}}}_Y^{\mathrm{num}}(s)/\mathbb{P}_{1,n}[h_A(x)]$ is asymptotically linear,

$$\widehat{\overline{\mathrm{McEE}}}_Y^{\mathrm{num}}(s)/\mathbb{P}_{1,n}[h_A(x)]-\mathrm{McEE}^{\mathrm{num}}_Y(s)/\mathbb{P}_1[h_A(x)]=\mathbb{P}_{1,n}\psi_{Y,s}(X,Y,\eta^{\mathrm{num}}_{Y,s})+o_p(n^{-1/2}),$$

with the influence function

$$\psi_{Y,s}(X,Y;\eta^{\mathrm{num}}_{Y,s})=\frac{1}{\mathbb{P}_1[h_A(x)]}\psi^{\mathrm{num}}_{Y,s}(X,Y;\eta^{\mathrm{num}}_{Y,s})-\frac{\mathrm{McEE}^{\mathrm{num}}_Y(s)}{(\mathbb{P}_1[h_A(x)])^2}h_A(x)$$

where $\psi_{Y,s}^{\text{num}}(X, Y; \eta_{Y,s}^{\text{num}})$ is defined in (21).

As a consequence, the estimator is asymptotically normal,

$$\sqrt{n}\left(\widehat{\text{McEE}}_Y(s) - \text{McEE}_Y(s)\right) \rightarrow_d N(0, \text{var}(\psi_{Y,s}(X, Y; \eta_{Y,s}^{\text{num}}))) \tag{34}$$

$\square$

We can estimate the variance from the data to construct valid tests, for example a z-test or a Wald test. Alternatively, we can use a multiplier bootstrap which bypasses estimating the variance.

### D.2. Detailed test of covariate shift

The form of the debiased estimator $\text{McEE}_X$ is similar to those in indirect effects literature and follow straightforwardly from the approach outlined in the previous section. Therefore, we omit derivations for asymptotic linearity. See Lemma G.2 in Singh et al. (2024) for an illlustration.

We require the following conditions for asymptotic linearity. For the density ratios $\pi_A, \pi_{s,A}$ to be well-defined, we need the support of covariates in the source domain to be larger than the support in the target domain. This can be ensured by restricting the target domain to the common support (Cai et al., 2023). Importantly, we require that only the product of estimation errors of the nuisance parameters is asymptotically negligible at $o_p(n^{-1/2})$. Therefore, even though the estimator involves fitting many models, debiasing relaxes the conditions required on their estimation errors.

**Condition D.4.** Assume the following holds,

- (consistency) Nuisance parameter estimates $\hat{Z}_0, \hat{Z}_{0,s}, \hat{\pi}_A, \hat{\pi}_{s,A}$ are consistent.
- (contiguity) $p_0(x) > 0$ whenever $p_1(x) > 0$.
- (product of errors in $\hat{Z}_0$ and $\hat{\pi}_A$) $\mathbb{P}_0(Z_0(x) - \hat{Z}_0(x))(\hat{\pi}_A(x) - \pi_A(x)) = o_p(n^{-1/2})$
- (product of errors in $\hat{Z}_{0,s}$ and $\hat{\pi}_{s,A}$) $\mathbb{P}_0(\hat{Z}_{0,s} - Z_{0,s})(\hat{\pi}_{s,A} - \pi_{s,A}) = o_p(n^{-1/2})$.

**Theorem D.5.** *Given Condition (D.4) holds, the estimator $\widehat{\text{McEE}}_X(s) = \widehat{\text{McEE}}_X^{\text{num}}(s)/\mathbb{P}_n[h_A(x)]$ defined in (17) follows a normal distribution asymptotically, centered at the estimand $\text{McEE}_X(s)$.*

## E. More details on related work

**Hypothesis tests for distribution shift.** Rabanser et al. (2019) compares multiple distribution shift detection methods and find that two-sample testing on learned representation works well. Kulinski et al. (2020) finds features that shift via conditional independence tests using a divergence measure based on score function. However, it requires specifying a density model, a challenging problem in high-dimensions, and does not provide inferential guarantees. Hindy et al. (2024) proposes a conformal prediction method to test for distribution shift over time. They focus on finding shifts in intermediate steps of a multi-step process instead of finding feature subsets. Much of the shift detection literature does not focus on finding shifts that significantly vary performance. In contrast, Panda et al. (2024) proposes a hypothesis test to find contributing features for a performance change. To determine whether a feature is important, they test whether the effect of adding the feature to the algorithm on its performance is the same across the two distributions. However, the specifics of how they add/remove features, essentially by zero-ing in the features, do not respect feature correlations. They do not show whether the tests have a controlled type-I error rate, which means the test could falsely identify many features as important.

**Hypothesis tests for unfairness.** One application of SHIFT is to test for disparities in an algorithm's performance (unfairness) by considering the source and target datasets to be data for the two protected groups. Hence, literature on auditing unfairness is relevant. Zahn et al. (2023) searches through the covariate space to find subgroups with high unfairness. The tests are limited to certain fairness notions such as demographic parity and equalized odds, and make parametric assumptions for inference since they use a $\chi^2$ test. We note that an extensive literature exists on multi-calibration or multi-accuracy which aims to train models that are accurate for all computationally-identifiable groups (Kearns et al., 2018; Hebert-Johnson et al., 2018; Kim et al., 2019). Apart from similarities in the focus on identifiable subgroups, the objectives of the work are different. Such methods are complementary to our testing approach and can be used if the tests suggest to retrain the model.

**Challenges in testing when performance decay is zero.** Prior works (Hines et al., 2023; Quintas-Martinez et al., 2024; Singh et al., 2024) on VI for treatment effects propose tests that suffer from undercoverage when the null hypothesis

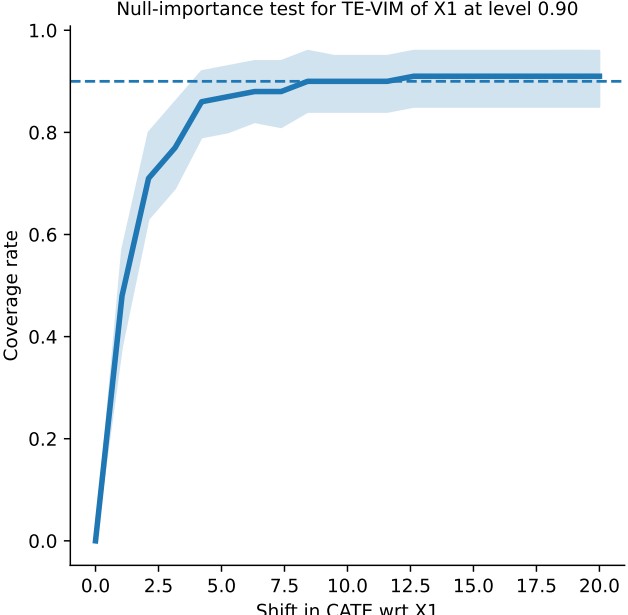

Figure 6: **Testing VIs at boundary of parameter space.** Existing methods for the related problem of explaining treatment effect heterogeneity (Hines et al., 2023; Quinzan et al., 2023; Williamson et al., 2021) do not give valid tests for importance of a variable. We observe that the coverage rate of confidence intervals for their test statistics (specifically, `TE-VIM`) severely decreases below the desired rate when the variable importance goes to zero.

holds (Figure 6). Therefore, they do not give valid tests for importance of a variable. We observe similar behavior for the comparator `TE-VIM` in experiments.

## F. Options for defining detailed shifts with respect to a subset

We discuss some alternatives to defining the detailed level tests.

**Detailed outcome shifts.** Another choice for the null hypothesis $H_{0,s}^{Y|X}$ is that

$$\sup_h E[Z_1(x) - Z_0(x) - Z_s^*(x_s)|x \in A] = 0$$

where $Z_s(x_s) = E_1[Z_1(x) - Z_0(x)|x_s]$.

This test, however, will fail to flag some subsets as important explanations. To illustrate, consider a simple example where the conditional loss function does not change with respect to a subset but still the hypothesis will be rejected.

**Example 1.** Assume the conditional loss functions for the domains are $Z_0(x) = \sigma(x_1 + 0.5x_2)$ and $Z_1(x) = \sigma(x_1 + x_2)$ for the sigmoid function $\sigma(x) = 1/(1 + exp(-x))$. Even though there is no change in loss functions with respect to $x_1$, the null hypothesis does not hold for $x_1$ since the difference of sigmoids is still a function of $x_1$. However, the target loss can be defined as a function of $x_2$ and $Z_0(x)$ alone, $Z_1(x) = \sigma(\sigma^{-1}(Z_0(x)) + 0.5x_2)$.

The unintuitive behavior in the example occurs because we explain the discrepancy between $Z_1$ and $Z_0$ by taking their difference as opposed to any other scale.

**Detailed covariate shifts.** A natural choice for the null hypothesis is to posit that $Z_0(x)P_1(x) = Z_0(x)P_0(x)$ for all $x$. However, this formulation would attribute performance change to covariate shifts in variables which are irrelevant to the loss function. Consider a simple example to illustrate the issue.

**Example 2.** Assume conditional loss and one of the two features are distributed the same in source and target, $Y|X = (x_1, x_2) \sim N(x_1, 1)$, $x_1 \sim N(0, 1)$ in $P_0$ and $P_1$. For simplicity, consider $\ell = Y$. Second feature has a covariate shift from

$x_2 \sim N(0, 1)$ in $P_0$ to $x_2 \sim N(1, 1)$ in $P_1$. That is, $P_1(x) \neq P_0(x)$. Since $Z_0(x) = Z_0(x_1)$ and $x_1$ does not shift, there is no performance change. Even though $x_2$ is irrelevant to the loss, the shift in $x_2$ will leads us to reject the null and attribute performance change to covariate shift.

As the example suggests, one way to address this undesired behavior is to focus only on shifts in variables important to the conditional loss $Z_0(x)$ instead of all variables $x$.

## G. Experiment details

Code to reproduce the experiments is at the link http://github.com/jjfeng/shift. We describe important details of the experiments and the implementation.

### G.1. Simulations

For simulation setups 2 and 3, we generate $n = 8000$ data points in both the source and target domains. We split datasets into equal halves for training and evaluation. For Setup 1, we reduce data points to $n = 2000$ for all methods since the kernel-based baselines take a long time. We split the sample in the ratio 20-80% for SHIFT to ensure more data for evaluation. The ML algorithm is a logistic regression model fitted on a separate sample of $n = 10000$ points from the source domain.

*Setup 1a* (Outcome shift in a subgroup): Define a subgroup $A = \{x \in \mathbb{R}^{10} | x \notin [-3.5, 3.5]\}$. $m_0 = \mu_1 = (0, 0, 0, 0, 0, 0, 0, 0, 0, 0), \Sigma_0 = \Sigma_1 = \text{diag}(2, 2, 2, 2, 2, 2, 2, 2, 2, 2), \phi_0 = (0.8, 0.5, 1, 0.1, 0.1, 0.1, 0.1, 0.1, 0.1, 0.1)$. The logit for target domain is $\phi_1 = (0.2, 0.5, 1, 0.1, 0.1, 0.1, 0.1, 0.1, 0.1, 0.1)$ when $x \in A$ and $\phi_0(x)$ otherwise. Accuracy overall is similar across source and target (drops by 0.5%). Since the baselines test for any difference in conditional outcome distribution, SHIFT similarly tests for $\tau = 0$. Given the subgroup $A$ is 7.8% of the data, SHIFT tests for minimum prevalence of $\epsilon = 0.05$.

*Setup 1b* (Covariate shift in a subgroup): Logits are same as Setup 1a $\phi_1 = \phi_0 = (0.8, 0.5, 1, 0.1, 0.1, 0.1, 0.1, 0.1, 0.1, 0.1)$. We vary the mean of the first covariate from 1 to 0 in the subgroup defined as $A = \{x \in \mathbb{R}^{10} | x_1 \notin [-4, 4]\}$. Covariance matrices are the same as Setup 1a. SHIFT tests for tolerance $\tau = 0$ and prevalence of $\epsilon = 0.05$.

*Setup 2* (Outcome shift only): $m_0 = \mu_1 = (0, 0, 0, 0), \Sigma_0 = \Sigma_1 = \text{diag}(2, 2, 2, 2)$, $\phi_0(x) = 0.8x_1 + 0.5x_2 + x_3 + 0.6x_4$ and $\phi_1(x) = 0.2x_1 + 0.4x_2 + x_3 + 0.6x_4$. Therefore, the outcome shifts with respect to both $X_1$ and $X_2$, but the shift in $X_2$ is minimal and below tolerance $\tau$. Accuracy of the ML algorithm by 5.9% from 83.8% in source domain to 77.9% in target domain. SHIFT tests for $\tau = 0.05, \epsilon = 0.05$.

*Setup 3* (Covariate shift only): $m_0 = (1, 0, 0, 1), \Sigma_0 = \text{diag}(2, 2, 2, 2)$ and $\mu_1 = (0, 0, 0, 0), \Sigma_1 = \text{diag}(1, 2, 2, 2)$. $\phi_0 = \phi_1 = 2.5x_1 + x_2 + 0.5x_3 + 0.1x_4$. Both $X_1$ and $X_4$ shift but $X_4$'s shift is very small and below tolerance $\tau$. Accuracy drops by 5.4% from 90.9% to 85.5%. SHIFT tests for $\tau = 0.02, \epsilon = 0.05$.

### G.2. Real-world case studies

**Background of case studies.** We chose the case studies to mirror the real-world application of the framework. They consist of settings where covariate or outcome shifts impact performance and domain experts do not know which shifts are detrimental. Such settings are highly prevalent in healthcare where ML performance varies widely across hospitals and time.

The first case study is based on a systematic analysis in Liu et al. (2023) that analyzed performance drops of an algorithm for predicting insurance coverage across different US states in the American Community Survey dataset. Among many state pairs, Liu et al. (2023) primarily found a large decay when transfering the algorithm from Nebraska to Louisiana. We decided to dive deeper into this analysis by identifying which subgroups were affected and why. SHIFT detected that people who are unemployed or whose parents are not in the labor force experience a large decay (Fig 3c). Since health insurance coverage is tied to employment in the US, and insurance rates and incomes differ between the states, such a decay is expected.

The readmission case study analyzes an algorithm to predict readmission that is trained on a well-resourced academic hospital and applied to a safety-net hospital. Since safety-net hospitals serve patients regardless of their ability to pay, their populations are quite different. SHIFT detected that patients with many emergency encounters experience a large decay (Fig 3d), which is expected because safety-net hospital patients seek care from emergency departments for very different

reasons than at academic hospitals. Thus, SHIFT helps detect subgroups in realistic benchmarks.

**Health insurance prediction.** We extract datasets for the two states from the 2018 yearly American Community Survey. It is available to download using the `folktables` package (Ding et al., 2021). Source (Nebraska) and target (Louisiana) domains have 3166 and 12000 points respectively, and 34 features. For the covariate shift tests, we filter out features that are uncorrelated with the loss function, resulting in a total of 18 features. Outcome shift tests keep all features. The features are categorized into 5 broad categories (Table 6). Outcome is whether the person has public health insurance. We train a multi-layer perceptron on separate datasets of 3166 points from source domain. Public health insurance rate (class prevalence) increases drastically from 19.9% to 40.4% in Louisiana. SHIFT tests for tolerance $\tau = 0.05$ and prevalence $\epsilon = 0.05$.

Table 6: **Featues for health insurance prediction.** Demo refers to demographics and Misc refers to Miscellaneous features.

| Feature name | Category | SHAP importance for ML algorithm |
|---|---|---|
| Sex | Demo | 0.072 |
| Citizenship status not a citizen | Demo | 0.010 |
| Citizenship status naturalized | Demo | 0.001 |
| Citizenship status born abroad | Demo | 0.001 |
| Citizenship status born in Puerto Rico, Guam | Demo | 0.000 |
| Citizenship status born in US | Demo | 0.003 |
| Race White | Demo | 0.003 |
| Never married or under 15 years | Demo | 0.006 |
| Divorced | Demo | 0.043 |
| Widowed | Demo | 0.030 |
| Married | Demo | 0.062 |
| Separated | Demo | 0.002 |
| Nativity | Demo | 0.021 |
| Ancestry | Demo | 0.023 |
| Age | Demo | 0.018 |
| Cognitive difficulty | Health | 0.016 |
| Vision difficulty | Health | 0.007 |
| Hearing difficulty | Health | 0.001 |
| Disability | Health | 0.074 |
| Gave birth to child in past 12 months | Misc | 0.006 |
| Military service | Health | 0.054 |
| Mobility status | Health | 0.052 |
| Employment status of parents | Health | 0.022 |
| Employment status armed | Employment | 0.000 |
| Employment status unemployed | Employment | 0.009 |
| Employment status partial employed | Employment | 0.001 |
| Employment status employed | Employment | 0.008 |
| Employment status not in labor force | Employment | 0.012 |
| Employment status partial armed | Employment | 0.000 |
| Total person's income | Employment | 0.244 |
| School at least high school or GED | Education | 0.000 |
| School at least bachelor | Education | 0.010 |
| Educational attainment | Education | 0.186 |
| School postgrad | Education | 0.001 |

**Readmission prediction.** We access electronic health records from an academic and a safety-net hospital and extract datasets for predicting readmission within 30-day of discharge for patients diagnosed with heart failure. Source (academic hospital) and target (safety-net hospital) domains have 7468 and 6515 points respectively, and 27 features. Covariate shift tests are run after filtering out features uncorrelated with the loss function, resulting in a total of 20 features. The features

belong to 4 broad categories (Table 7). We train a random forest on a separate dataset of 22403 points from source domain. Readmissions increase considerably from 12.4% to 18.5% in the safety-net hospital. SHIFT tests for tolerance $\tau = 0.02$ and prevalence $\epsilon = 0.05$.

Table 7: **Features for readmission prediction.** Demo refers to demographic variables.

| Feature name | Category | SHAP importance for ML algorithm |
|---|---|---|
| Num ED encounters | Encounters | 0.499 |
| Firstrace Decline/Other/Unknown | Demo | 0.005 |
| Sex Female | Demo | 0.004 |
| First race White | Demo | 0.000 |
| First race Native Hawaiian or Other Pacific Islander | Demo | 0.000 |
| Sex Male/Other | Demo | 0.004 |
| First race Black or African American | Demo | 0.007 |
| First race Asian | Demo | 0.003 |
| Age | Demo | 0.026 |
| First race Native American or Alaska Native | Demo | 0.000 |
| Vital Weight/Scale | Vitals | 0.003 |
| Vital Pulse | Vitals | 0.007 |
| Labs Calcium, total, Serum / Plasma | Labs | 0.001 |
| Labs Hemoglobin | Labs | 0.233 |
| Labs Potassium, Serum / Plasma | Labs | 0.032 |
| Labs MCV | Labs | 0.004 |
| Labs Chloride, Serum / Plasma | Labs | 0.003 |
| Labs WBC Count | Labs | 0.015 |
| Labs Lactate, whole blood | Labs | 0.020 |
| Labs Carbon Dioxide, Total | Labs | 0.000 |
| Labs eGFR - low estimate | Labs | 0.021 |
| Labs eGFR - high estimate | Labs | 0.023 |
| Labs Phosphorus, Serum / Plasma | Labs | 0.016 |
| Labs Sodium, Serum / Plasma | Labs | 0.022 |
| Labs Magnesium, Serum / Plasma | Labs | 0.022 |
| Labs Creatinine | Labs | 0.017 |
| Labs Anion Gap | Labs | 0.011 |

**Model update experiment.** We demonstrate the actionability of the SHIFT results on health insurance prediction. The detailed outcome shift test in Figure 3c detects a subgroup affected by outcome shifts and identifies demographic variables as a potential explanation. Based on this, SHIFT allows a targeted update to mitigate the decay in the detected subgroup. To address outcome shift, we fine-tune the ML algorithm with respect to demographic variables by fitting a new model that takes original algorithm's predictions and demographic variables to predict the outcome in target data. The updated algorithm applies the new model only for the detected subgroup and defaults to the original model otherwise. Thus, the targeted update addresses the decay in affected subgroup while keeping the algorithm behavior the same for everyone else.

We compare the targeted update to a standard practice in response to distribution shifts that is to retrain the model on all features and use it for everyone. We call it a non-targeted update. Additionally, we update the model on only the employment-related features identified by the TE-VIM method. Table 3 reports the AUC of the targeted and the two non-targeted updates. For the update, we take 2000 points from target domain held out from earlier tests and fit an MLP model. We evaluate the updates (on subgroups) in another held-out 2440 points from target domain. We observe that the targeted update achieves around the same performance on the detected subgroup as the non-targeted one. However, the non-targeted have the unintended effect of reducing the performance on another subgroup where the original model performed better. We find the other subgroup affected by the non-targeted updates by again applying SHIFT. Hence, the experiment demonstrates that SHIFT can help to adapt ML algorithms to new settings while making minimal changes to them.

# H. Implementation details

**Estimation of nuisance parameters.** We use sample-splitting to estimate all nuisance parameters on 50% of the source and target domain data and evaluating the test statistics on the remaining data. We use implementations provided in `scikit-learn` package to fit ML models (Pedregosa et al., 2011). For outcome models, we search through a grid of estimators and their hyperparameters through cross-validation, which includes logistic regression, random forest, and gradient boosting classifiers. Density models are chosen from logistic regression and calibrated random forest. We clip the density ratios by restricting predicted probabilities from the density models between $[10^{-3}, 1 - 10^{-3}]$ to avoid high variance in our McEE estimates. We bin the outcome shift $\mu_{\text{bin}}$ into $B = 40$ equally-spaced intervals in $[0, 1]$ for detailed outcome shifts in all experiments.

**Setting testing hyperparameters $\tau$ and $\epsilon$.** The hypotheses require specifying two hyperparameters, namely, minimum subgroup size $\epsilon$ and minimum shift magnitude $\tau$. Both are intuitive and can be set by anyone, but they are domain-specific. The $\tau$ specifies the smallest performance change that is deemed to be meaningful to detect. For safety-critical settings like healthcare, even a 1-2 % change might be meaningful. Accuracy changes below $\tau$ are part of the null hypotheses and will not be flagged, hence allowing the domain expert to ignore very small changes.

The $\epsilon$ allows us to only consider shifts that affect subgroups of at least size $\epsilon$. If the prevalence of the subgroup experiencing performance change goes below $\epsilon$, the null hypothesis would be true and this tiny subgroup would no longer be of interest. In general, the power of SHIFT decreases as the prevalence of the shifting subgroup decreases. Existing tests also decrease in power but they set the minimum threshold to zero, stating that all shifts are of practical interest and yet have limited power to detect them. We show in experiments (Figures 8 and 7) that SHIFT can detect changes in datasets with as low as 500 data points ($\epsilon = 5\%$ would mean subgroups of 25 data points), but we recommend setting $\epsilon$ to have at least 100 data points to have decent power.

Together, $\tau$ and $\epsilon$ ensure that the tests are practical and reduce alarm fatigue caused by detecting negligible shifts.

**Computational complexity.** SHIFT runs in under 10 minutes for the real-world datasets with around 10,000 points. The bulk of the computation is dedicated to fitting the nuisance models, so the runtime is $O(V)$ where $V$ is the number of cross-validation folds. Moreover, fitting these nuisance models is easily parallelizable.

**Visualizing the fitted detectors.** We explain the detectors by fitting decision sets using the MLIC package (Ghosh et al., 2022). Decision sets output propositional logic statements on the features, such as (If $X_1 > 0$ AND $X_2 < 1$), to classify points into detected and not detected. We fit decision sets to classify detected subgroups in either source or target data, and report the sets for the detected class in Figures 3a, 3b, 3c, and 3d.

**Comparators.** We give details for the kernel-based comparators for the aggregate tests.

Aggregate tests: For outcome shift tests, KCI tests for $D \perp \ell | X$. MMD typically is used to test unconditional independence between two variables. We repurpose MMD for conditional outcome shifts by testing $D \perp (\ell, X)$. Since the joint distribution factorizes into $p(\ell, X, D) = p(\ell | X, D) p(X | D)$ and we know that covariate distribution does not vary in Setup 1, the null hypothesis is equivalent to $D \perp \ell | X$.

For the comparators of the detailed tests, $X_s$ is a good explanation when the null is rejected whereas it is likely to be a good explanation when we fail to reject.

Detailed outcome: Tests are set up such that the subset explains the outcome shift if we fail to reject the null for (a) and (d), and if we reject the null in the case of (b) and (c).

Detailed covariate: Tests are set up such that the subset explains the covariate shift if we fail to reject the null for (c), and if we reject the null in the case of (a) and (b).

# I. Additional simulation results

We validate type-I and power properties of the tests by repeating them on $N = 50$ random draws of the dataset with varying sample size and reporting the rejection rates.

Recall that Setup 2 only has conditional outcome shifts with respect to variables $X_1$ and $X_2$. Aggregate tests reject the outcome shift test with power tending to 1 as sample size increases (Figure 8). Rejection rate for agg X-test, which corresponds to an estimate of type-I error, is within $\alpha = 0.05$. Since $X_1$ is the variable with the most shift, the test for

subset $X_s = \{X_1\}$ has a low rejection rate. The test for subset $\{X_1, X_2\}$ achieves type-I error control. Alone neither of $X_2$, $X_3$, and $X_4$ explains the shift, thus, they are correctly rejected. The rejection rates tend to 1 even with $n = 2000$ samples. Figure 9a shows that none of the comparators flags the shifting subset.

The advantage of double/debiased inference is evident from the results for Setup 3 (Figure 7). Recall that covariate distribution shifts only for $X_1$ and $X_4$ in Setup 3. For the aggregate tests, plug-in rejects agg Y-test even when there is no outcome shift. On the other hand, one-step achieves type-I error control. Similarly, we observe that detailed test for subset $\{X_1, X_4\}$ achieves type-I error control. However, the type-I error control comes at the cost of lower power in case of the remaining subsets. Thus, the one-step tests are likely to flag more subsets as candidates than needed to explain the performance shift. It could be desirable to have such conservative tests if we do not want to miss potential ways to fix the performance shift. Figure 9b shows that `KCI` identifies the shifting subset correctly while other comparators testing subsets either in principle (`KS` is for univariate samples) or in their implementation (`Score`).

## J. Additional case study

Although SHIFT is primarily designed for tabular data, its aggregate-level tests are suitable for analyzing unstructured data; its detailed-level tests can also be used, if one has prespecified concepts. As an example, we apply SHIFT to the CivilComments dataset (Koh et al., 2021), which contains comments on online articles and are judged to be toxic or not. We consider a DistilBERT-base-uncased model (Sanh et al., 2019) fine-tuned to classify toxic comments. Given the 768-dimensional embeddings from this BERT model, we can apply SHIFT to understand differences in accuracy when classifying comments that mention the female gender (target domain) versus the remaining (source). Accuracy of the model drops by 1.3% in the target. Results from SHIFT's aggregate-level test find evidence for covariate shift, i.e. there exists a subgroup of size $\geq 5\%$ that experiences an accuracy drop greater than $5\%$ due to covariate shift (Table 8).

Table 8: **Results on a high-dimensional text dataset.** We report p-values from the aggregate tests on CivilComments data which consists of internet comments and their toxicity labels. SHIFT finds evidence for accuracy drop due to covariate shift.

| Aggregate test | p-value |
| --- | --- |
| Covariate shift | 0.00 |
| Outcome shift | 0.83 |

To run detailed-level tests in SHIFT, we require variables to be interpretable. Given unstructured data, one solution is to combine SHIFT with concept bottleneck models (Koh et al., 2020). We note that another solution, if one does not need statistical inference at the detailed level, is to simply analyze differences between the comments from the detected subgroup from SHIFT in the source and target domains. Using a combination of GPT-4o (OpenAI et al., 2024) and manual review, we found that in the subgroup where the toxicity classifier experienced performance decay at the target domain, the comments tended to discuss politics, society, race, and identity more. This shift in topics may explain the performance drop. For instance, the combination of female references with discussions of race and political ideology might compound biases that the classifier has inadvertently learned.

Finally, we note that although the proofs for SHIFT's validity require sufficiently fast estimation rates for the nuisance parameters which tend to slow down as dimensionality increases, estimation rates for nuisance parameters may still be sufficiently fast if these parameters are sparse (e.g. (Wager & Walther, 2015; Belloni & Chernozhukov, 2011)).

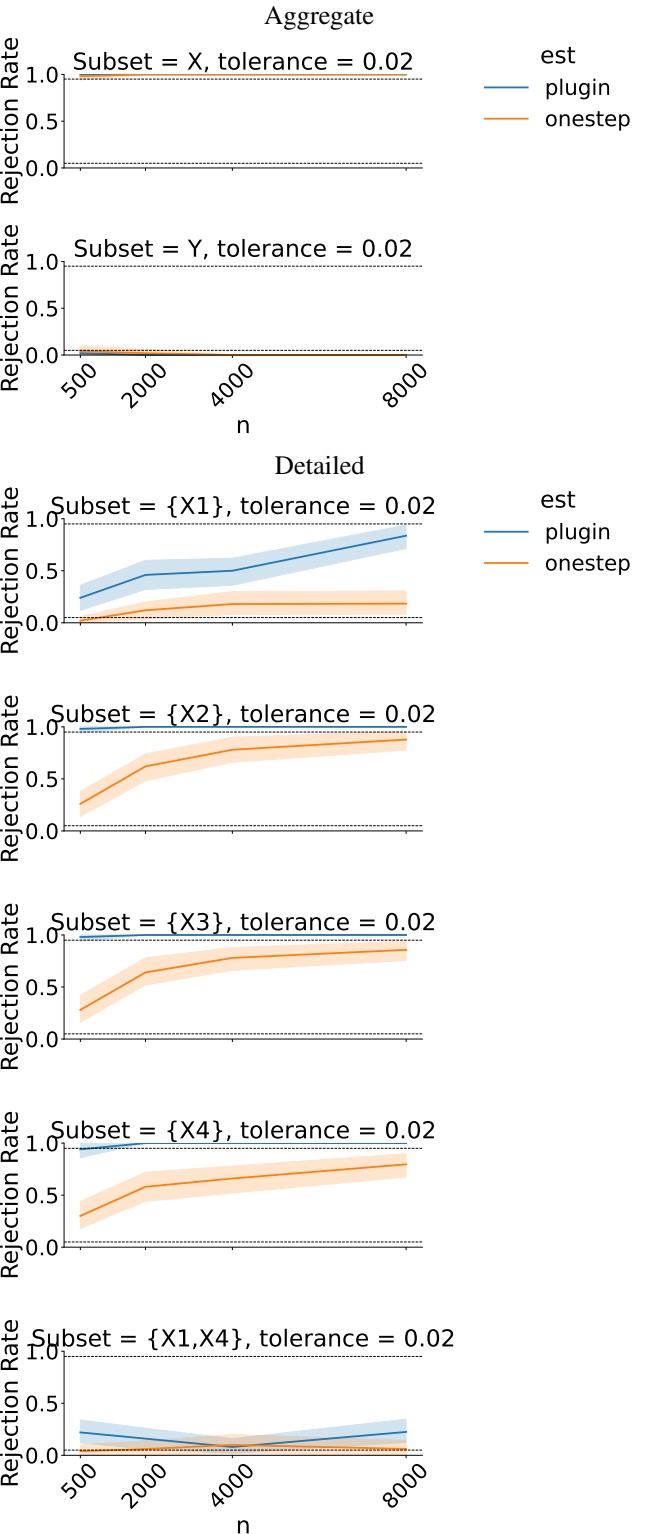

Figure 7: **Covariate shift setup.** Covariate test, tolerance $\tau = 0.0$ and prevalence $\epsilon = 0.05$.

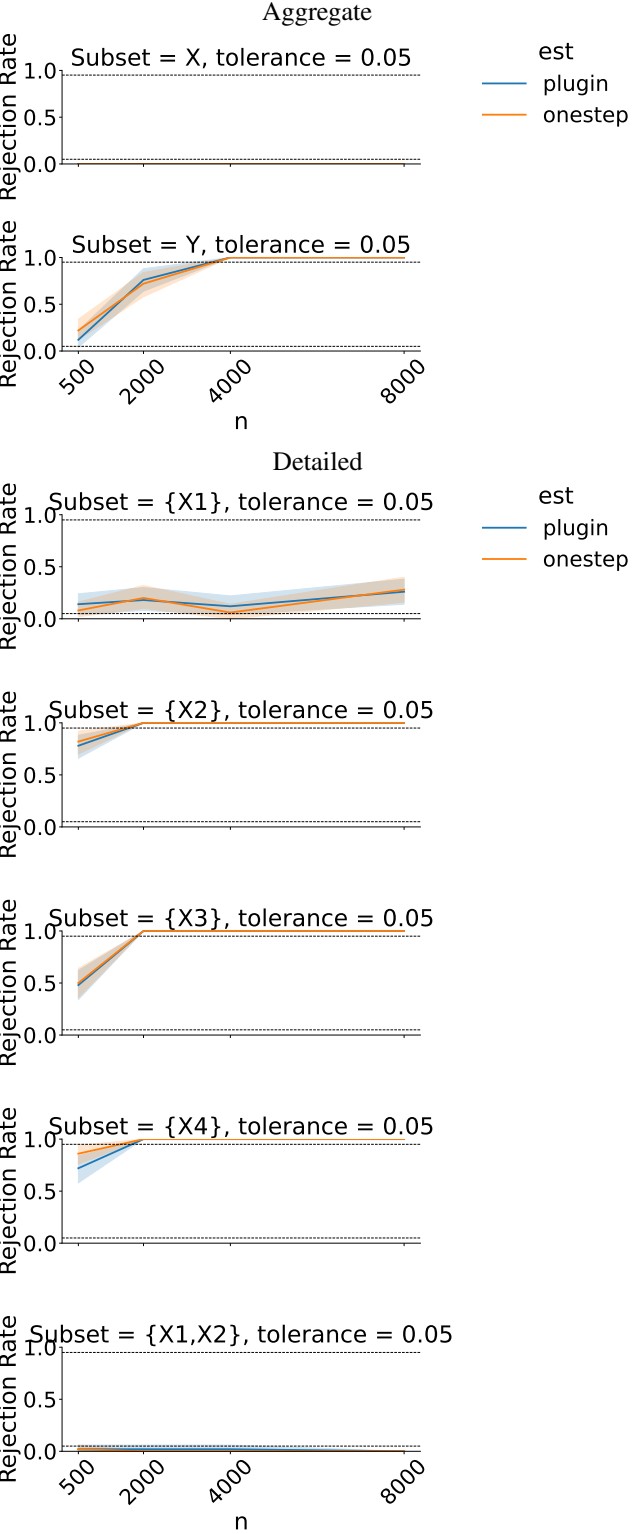

Figure 8: **Outcome shift setup.** Outcome test, tolerance $\tau = 0.05$ and prevalence $\epsilon = 0.05$.

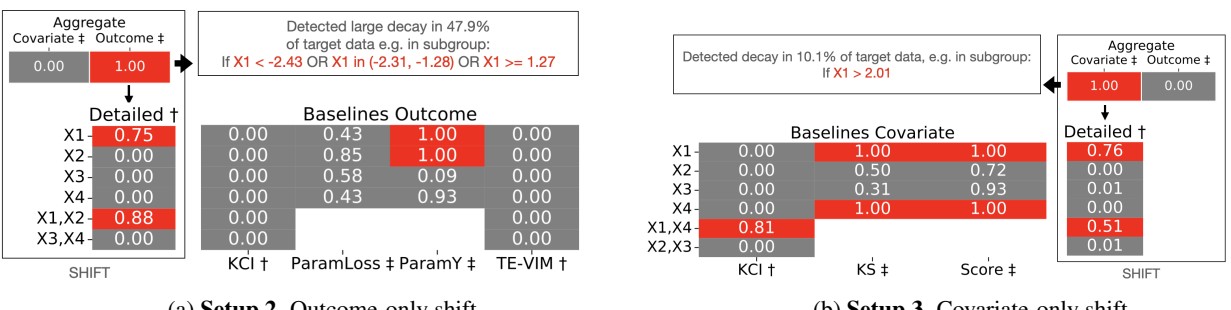

(a) **Setup 2.** Outcome-only shift

(b) **Setup 3.** Covariate-only shift

Figure 9: Detailed tests for variable subsets in simulation setups.

