# OpenReview forum: ""Who experiences large model decay and why?" A Hierarchical Framework for Diagnosing Heterogeneous Performance Drift"
_ICML.cc/2025/Conference — ICML 2025 poster_

### Official Review · Reviewer_buFa · 2025-03-14

**Overall Recommendation:** 3

**Summary:**

This paper proposes a nonparametric Subgroup-scanning Hierarchical Inference Framework for performance drifT (SHIFT) to use hypothesis testing for drift diagnosis. The SHIFT first decides if any subgroup experiences significant performance decay from drift, then checks the specific shift that explains the decay. In this way, SHIFT enables the explainable detection of subgroup shifts.

## update after rebuttal
The rebuttal from the authors solved my concerns and provided insightful experiment and discussion. I decide to keep my original positive rating.

**Claims And Evidence:**

The claims are successfully supported by experiments on tabular data.

**Essential References Not Discussed:**

I didn't identify any such references.

**Experimental Designs Or Analyses:**

The experiments are extensive and support the proposed claims.

**Methods And Evaluation Criteria:**

The proposed methods and evaluation criteria make sense under the problem setting and tabular data.

**Other Comments Or Suggestions:**

NaN

**Other Strengths And Weaknesses:**

Strengths:

This paper is well-written and easy to follow. The work proposes a framework to detect the subgroup with performance decay and identify the key variables for the performance decay, which is explainable and has great potential in practical applications. The application of SHIFT in practice is also well-discussed in this paper.

Weaknesses:

This paper mainly focuses on the experiments with tabular data. Despite the conclusion mentions it can be applied on image/text data, it might bring new challenges, such as computational cost and the categorization of subgroups.

**Questions For Authors:**

1. How is the efficiency or computational cost of the hypothesis testing in SHIFT?

2. Will there be new challenges when extending SHIFT to image/text data? If so, is it possible to address these challenges?

3. SHIFT requires domain experts to select the minimum subgroup size and minimum shift magnitude and how sensitive is it to the choices of the hyperparameters?

**Relation To Broader Scientific Literature:**

Previous work proposes to decompose the average performance drop within an identified subgroup into covariate shift and outcome shift. This work aims to further identify the related input variables. In addition, the prior work on drift diagnosis is primarily based on the estimation, while this work uses hypothesis testing.

**Theoretical Claims:**

The theoretical claims focus on the formulation of the hypothesis testing.

---

> ### Author Rebuttal · Authors · 2025-04-01
>
> We thank the reviewer for their constructive feedback and are glad to know that they found the work to be well-written and to have **great potential in practical applications**. Indeed, SHIFT addresses a **critical and very practical question**: when performance of an ML algorithm drops in a new application context, which subgroups are adversely affected and why? Such settings are prevalent in healthcare, where ML algorithms differ in performance widely across demographic groups, time, and geographies [Finlayson et al. 2021].
>
> Discovering the subgroups experiencing performance drop is **highly important to catch hidden failures of the algorithm and to develop targeted fixes** to the algorithm for the affected subgroups without sacrificing its performance elsewhere. Current methods to study performance drops either do not focus on subgroup-level performance or do not quantify uncertainty in the discovered subgroups. SHIFT addresses this methodological gap via statistically principled and computationally scalable methods. Furthermore, we believe SHIFT provides a **solid theoretical foundation** on which future work can build on, as discussed below.
>
> ---
>
> **Extending SHIFT to image/text data** Although SHIFT is primarily designed for tabular data, its aggregate-level tests are suitable for analyzing unstructured data; its detailed-level tests can also be used, if one has prespecified concepts. In the revised manuscript, we will include applications of SHIFT to both text and image datasets. As an example, we have applied SHIFT to the CivilComments dataset [Koh et al. 2021], which contains comments on online articles and are judged to be toxic or not. We consider a DistilBERT-base-uncased model fine-tuned to classify toxic comments. Given the **768-dimensional** embeddings from this BERT model, we can apply SHIFT to understand differences in accuracy when classifying comments that mention the female gender (target domain) versus the remaining (source). Accuracy of the model drops by 1.3\% in the target. Results from SHIFT's aggregate-level test find evidence for covariate shift, i.e. there exists a subgroup of size $\ge$ 5\% that experiences an accuracy drop greater than 5\% due to covariate shift.
>
> | Test            | p-value |
> |-----------------|---------|
> | Covariate shift | 0.00    |
> | Outcome shift   | 0.83    |
>
> To run detailed-level tests in SHIFT, we require variables to be interpretable. Given unstructured data, one solution is to combine SHIFT with concept bottleneck models [Koh et al. 2020]. We will include such an example in the revised paper. We note that another solution, if one does not need statistical inference at the detailed level, is to simply analyze differences between the comments from the detected subgroup from SHIFT in the source and target domains. Using a combination of GPT-4o and manual review, we found that in the subgroup where the toxicity classifier experienced performance decay at the target domain, the comments tended to discuss politics, society, race, and identity more. This shift in topics may explain the performance drop. For instance, the combination of female references with discussions of race and political ideology might compound biases that the classifier has inadvertently learned.
>
> ---
>
> **Computational cost** SHIFT is very **fast**. SHIFT runs in under 10 minutes for the real-world datasets with around 10,000 points. The bulk of the computation is just fitting the nuisance models, so the runtime is just O(V) where V is just the number of cross-validation folds. Moreover, fitting these nuisance models is easily parallelizable.
>
> ---
>
> **Sensitivity to test parameters** We find that SHIFT is not very sensitive to the two test parameters: minimum subgroup size and shift magnitude. Moreover, these two parameters are very **intuitive**. While one could have a domain expert set them, they are intuitive parameters that can be easily selected by anyone and should simply reflect one's own tolerance for performance drift. We will include guidance on setting the parameters in the two case studies.
>
> We hope that the responses address your concerns.
>
> ---
>
> [1.] Koh et al. WILDS: A Benchmark of in-the-Wild Distribution Shifts. ICML 2021

---

> > ### Comment · Reviewer_buFa · 2025-04-04
> >
> > Thank you for the rebuttal from the authors. The application of SHIFT on CivilComments looks promising and the discussion has the potential to inspire future research. I will keep my positive rating and hope the authors could incorporate the new results in the draft.

---

### Official Review · Reviewer_g6ej · 2025-03-14

**Overall Recommendation:** 3

**Summary:**

This paper introduces SHIFT, a hierarchical hypothesis-testing framework designed to identify subgroups experiencing significant performance degradation in machine learning models due to distribution shifts. SHIFT first tests for the presence of large performance decay due to aggregate covariate and outcome shifts, and subsequently identifies specific subsets of input variables responsible for the observed decay. Unlike previous approaches, SHIFT does not rely on strong parametric assumptions or detailed causal knowledge, making it suitable for scenarios with limited data. For experiments, SHIFT is validated at the simulation level and also for shifts in real-world data.

**Claims And Evidence:**

The claimed are backed by theoretical proofs. Empirically, it was shown via experiments (both simulated and real) that SHIFT can identify relevant shifts and improve subgroup accuracy.

The paper could be further strengthened by demonstrating SHIFT's robustness across a wider array of distribution shifts such as  high-dimensional, non-tabular data. Additionally, observations under extreme sparsity or very small sample scenarios would be informative.

**Essential References Not Discussed:**

It could benefit the paper if the WILDS dataset (Koh et al., 2021), real-world distribution shift benchmark, is included.

**Experimental Designs Or Analyses:**

Strengths:

- Clear ground-truth validation of subgroup detection and shift attribution via simulations.
- Also demonstrates practical utility via real-world studies, with SHIFT-driven fixes improving performance.

Weaknesses:

- Scalability - no discussion of runtime or feasibility in high-dimensional settings.
- SHIFT focuses on one affected subgroup per shift type, but iterative subgroup discovery is not explored.

**Methods And Evaluation Criteria:**

The proposed method’s two-stage testing approach makes sense. Also, the experimental evaluation is sound, using both controlled simulation and real world case studies encompassing realistic shifts (insurance across states, hospital readmission across institutions). If possible, additional real world studies regarding a different domain would be very informative, but it could be hard to conduct such experiments.

**Other Comments Or Suggestions:**

Currently, I have no other comments or suggestions.

**Other Strengths And Weaknesses:**

My thoughts on the strengths and weakness of the paper are discussed above.

**Questions For Authors:**

Currently, I have no other questions. If it is possible, I wish to confer with other reviewers regarding the theoretical aspects of the manuscript.

**Relation To Broader Scientific Literature:**

The proposed method is closely related to distribution shifts, robust machine learning, bias and fairness in machine learning. Although the paper primarily focuses on tabular data, these concepts are also important for large scale models (if the proposed method could be effectively scaled so such magnitudes).

**Theoretical Claims:**

As I am not currently well-versed in the theoretical aspects of the related literature, I must defer the analyses and verifications of the theoretical claims of the paper to other reviewers.

---

> ### Author Rebuttal · Authors · 2025-04-01
>
> We thank the reviewer for their positive feedback appreciating the practical utility of the methods and for providing helpful suggestions on additional benchmarks. Indeed, SHIFT addresses a **critical and very practical question**: when performance of an ML algorithm drops in a new application context, which subgroups are adversely affected and why? Such settings are prevalent in healthcare, where ML algorithms differ in performance widely across demographic groups, time, and geographies [Finlayson et al. 2021].
>
> Discovering the subgroups experiencing performance drop is **highly important to catch hidden failures of the algorithm and to develop targeted fixes** to the algorithm for the affected subgroups without sacrificing its performance elsewhere. Current methods to study performance drops either do not focus on subgroup-level performance or do not quantify uncertainty in the discovered subgroups. SHIFT addresses this methodological gap via statistically principled and computationally scalable methods. Furthermore, we believe SHIFT provides a **solid theoretical foundation** on which future work can build on, as discussed below.
>
> ---
>
> **Computational complexity** SHIFT is very **fast**. SHIFT runs in under 10 minutes for the real-world datasets with around 10,000 points. The bulk of the computation is just fitting the nuisance models, so the runtime is just O(V) where V is just the number of cross-validation folds. Moreover, fitting these nuisance models is easily parallelizable.
>
> ---
>
> **Extensions to high-dimensional, non-tabular data e.g. WILDS** Although SHIFT is primarily designed for tabular data, its aggregate-level tests are suitable for analyzing unstructured data; its detailed-level tests can also be used, if one has prespecified concepts. In the revised manuscript, we will include applications of SHIFT to both text and image datasets. As an example, we have applied SHIFT to the CivilComments dataset [Koh et al. 2021], which contains comments on online articles and are judged to be toxic or not. We consider a DistilBERT-base-uncased model fine-tuned to classify toxic comments. Given the **768-dimensional** embeddings from this BERT model, we can apply SHIFT to understand differences in accuracy when classifying comments that mention the female gender (target domain) versus the remaining (source). Accuracy of the model drops by 1.3\% in the target. Results from SHIFT's aggregate-level test find evidence for covariate shift, i.e. there exists a subgroup of size $\ge$ 5\% that experiences an accuracy drop greater than 5\% due to covariate shift.
>
> | Test            | p-value |
> |-----------------|---------|
> | Covariate shift | 0.00    |
> | Outcome shift   | 0.83    |
>
> To run detailed-level tests in SHIFT, we require variables to be interpretable. Given unstructured data, one solution is to combine SHIFT with concept bottleneck models [Koh et al. 2020]. We will include such an example in the revised paper. We note that another solution, if one does not need statistical inference at the detailed level, is to simply analyze differences between the comments from the detected subgroup from SHIFT in the source and target domains. Using a combination of GPT-4o and manual review, we found that in the subgroup where the toxicity classifier experienced performance decay at the target domain, the comments tended to discuss politics, society, race, and identity more. This shift in topics may explain the performance drop. For instance, the combination of female references with discussions of race and political ideology might compound biases that the classifier has inadvertently learned.
>
> ---
>
> **Observations under very small sample scenarios** SHIFT performs well at small sample size. Please refer to Fig 6, 7 in Appendix, where we show that SHIFT maintains the specified type-I error rate and has good power as sample size decreases to 500.
>
> ---
>
> **Iterative subgroup discovery is not explored** Our tests provide evidence that there is an affected subgroup with statistical significance. We can then use existing subgroup discovery methods to explore all subgroups [Eyuboglu et al. 2022, d'Eon et al. 2022]. Such methods, thus, are complementary to ours but do not provide statistical significance.
>
> ---
>
> [1.] Koh et al. WILDS: A Benchmark of in-the-Wild Distribution Shifts. ICML 2021

---

> > ### Comment · Reviewer_g6ej · 2025-04-08
> >
> > Thank you for the detailed response. My raised concerns have been resolved.
> >
> > However, I wish to maintain my current positive score, as my evaluations are mostly based on the empirical side of the paper. If possible, I wish to defer the evaluations on the theoretical side to other reviewers.

---

### Official Review · Reviewer_JHtq · 2025-03-14

**Overall Recommendation:** 2

**Summary:**

The paper titled "Who experiences large model decay and why?" introduces a hierarchical framework called SHIFT (Subgroup-scanning Hierarchical Inference Framework for performance drifT) to diagnose heterogeneous performance drift in machine learning models. The framework aims to identify subgroups that experience significant performance decay due to covariate or outcome shifts and provides detailed explanations for these shifts. The goal is to enable targeted corrective actions that mitigate decay for the most affected subgroups.

**Claims And Evidence:**

The authors claim that existing methods do not provide detailed insights into subgroup-specific performance decay. SHIFT is proposed as a solution to identify and explain large performance decay in subgroups. The paper provides evidence through simulations and real-world experiments, demonstrating that SHIFT can identify relevant shifts and guide model corrections effectively.

**Essential References Not Discussed:**

NA

**Experimental Designs Or Analyses:**

The experiments include simulations and real-world case studies. Simulations vary the type and degree of shifts, ML algorithms, and data sizes to validate SHIFT's performance. Real-world case studies involve health insurance prediction across states and readmission prediction across hospitals. The experiments demonstrate SHIFT's ability to identify relevant shifts and guide targeted model updates.

**Methods And Evaluation Criteria:**

SHIFT is a two-stage hypothesis testing framework. The first stage identifies subgroups with large performance decay due to aggregate covariate or outcome shifts. The second stage provides detailed explanations by testing variable(subset)-specific shifts. The evaluation criteria include the ability to detect meaningful performance decay and provide valid statistical inference without strong assumptions.

**Other Comments Or Suggestions:**

The paper could benefit from a more detailed discussion on the computational complexity of the framework and potential strategies for efficient implementation. Additionally, providing more examples of real-world applications and their outcomes could enhance the practical relevance of the framework.

**Other Strengths And Weaknesses:**

Strengths of the paper include the novel hierarchical framework for diagnosing performance drift, the ability to provide detailed explanations for subgroup-specific shifts, and the use of hypothesis testing for valid statistical inference.

Weaknesses may include the complexity of the framework and the potential challenges in implementing it in practice. The reliance on domain experts to set parameters like minimum subgroup size and shift magnitude may also be a limitation.

**Questions For Authors:**

How does SHIFT handle cases where multiple subgroups experience overlapping shifts?
Can SHIFT be extended to handle unstructured data like images or text, and if so, how?
What are the computational requirements for implementing SHIFT in large-scale datasets?
How sensitive is SHIFT to the choice of parameters like minimum subgroup size and shift magnitude?
Are there any plans to make the SHIFT framework available as an open-source tool for broader use?

**Relation To Broader Scientific Literature:**

NA

**Theoretical Claims:**

The paper claims that SHIFT provides valid statistical inference through hypothesis testing, even with limited data. It does not rely on strong assumptions like knowledge of the true causal graph or large datasets. The theoretical properties of the framework are supported by asymptotic normality and controlled Type I error rates.

---

> ### Author Rebuttal · Authors · 2025-04-01
>
> We thank the reviewer for their comments and are heartened to hear they appreciate the novelty of SHIFT and its ability to provide detailed explanations for subgroup-specific shifts. Indeed, SHIFT addresses a **critical and very practical question**: when performance of an ML algorithm drops in a new application context, which subgroups are adversely affected and why? Such settings are prevalent in healthcare, where ML algorithms differ in performance widely across demographic groups, time, and geographies [Finlayson et al. 2021].
>
> Discovering the subgroups experiencing performance drop is **highly important to catch hidden failures of the algorithm and to develop targeted fixes** to the algorithm for the affected subgroups without sacrificing its performance elsewhere. Current methods to study performance drops either do not focus on subgroup-level performance or do not quantify uncertainty in the discovered subgroups. SHIFT addresses this methodological gap via **statistically principled and computationally scalable** methods. Furthermore, we believe SHIFT provides a **solid theoretical foundation** on which future work can build on, as discussed below.
>
> ---
>
> **Extensions to unstructured data**: Although SHIFT is primarily designed for tabular data, its aggregate-level tests are suitable for analyzing unstructured data; its detailed-level tests can also be used, if one has prespecified concepts. In the revised manuscript, we will include applications of SHIFT to both text and image datasets.
>
> As an example, we have applied SHIFT to the CivilComments dataset [Koh et al. 2021], which contains comments on online articles and are judged to be toxic or not. We consider a DistilBERT-base-uncased model fine-tuned to classify toxic comments. Given the **768-dimensional** embeddings from this BERT model, we can apply SHIFT to understand differences in accuracy when classifying comments that mention the female gender (target domain) versus the remaining (source). Accuracy of the model drops by 1.3\% in the target. Results from SHIFT's aggregate-level test find evidence for covariate shift, i.e. there exists a subgroup of size $\ge$ 5\% that experiences an accuracy drop greater than 5\% due to covariate shift.
>
> | Test            | p-value |
> |-----------------|---------|
> | Covariate shift | 0.00    |
> | Outcome shift   | 0.83    |
>
> To run detailed-level tests in SHIFT, we require variables to be interpretable. Given unstructured data, one solution is to combine SHIFT with concept bottleneck models [Koh et al. 2020]. We will include such an example in the revised paper. We note that another solution, if one does not need statistical inference at the detailed level, is to simply analyze differences between the comments from the detected subgroup from SHIFT in the source and target domains. Using a combination of GPT-4o and manual review, we found that in the subgroup where the toxicity classifier experienced performance decay at the target domain, the comments tended to discuss politics, society, race, and identity more. This shift in topics may explain the performance drop. For instance, the combination of female references with discussions of race and political ideology might compound biases that the classifier has inadvertently learned.
>
> ---
>
> **Computational complexity** SHIFT is very **fast**. SHIFT runs in under 10 minutes for the real-world datasets with around 10,000 points. The bulk of the computation is just fitting the nuisance models, so the runtime is just O(V) where V is just the number of cross-validation folds. Moreover, fitting these nuisance models is easily parallelizable.
>
> ---
>
> **Complexity in practice and open-source tools** We plan to publish an **open-source Python package** for running SHIFT that will provide simple and intuitive APIs. That way, users of the package can run SHIFT with just one or two lines of code.
>
> ---
>
> **Multiple subgroups experience overlapping shifts** SHIFT is designed to handle situations where the subgroup experiencing covariate shift does or does not overlap with the subgroup experiencing outcome shift. When there are multiple subgroups experiencing covariate shift, SHIFT groups them together into one large subgroup and performs an omnibus test. SHIFT handles multiple subgroups experiencing outcome shifts similarly.
>
> ---
>
> **Sensitivity to test parameters** We find that SHIFT is not very sensitive to the two test parameters: minimum subgroup size and shift magnitude. Moreover, these two parameters are very **intuitive**. While one could have a domain expert set them, they are intuitive parameters that can be easily selected by anyone and should simply reflect one's own tolerance for performance drift. We will include guidance on setting the parameters in the two case studies.
>
> We hope that our responses address the reviewer's concerns and encourage them to reconsider their score.
>
> ---
>
> [1.] Koh et al. WILDS: A Benchmark of in-the-Wild Distribution Shifts. ICML 2021

---

### Official Review · Reviewer_GEzi · 2025-03-16

**Overall Recommendation:** 3

**Summary:**

This paper proposes a method (SHIFT) for diagnosing performance drift in machine learning models that are transferred from a “source” to a “target” domain. Specifically, it aims to identify where (i.e., in which subgroups) a model’s performance decays the most and how such decay arises, distinguishing between subgroup-specific covariate shifts versus outcome shifts. By framing these questions as hierarchical hypothesis tests and using sample splitting plus flexible ML estimators, SHIFT produces valid inferences (i.e., with Type I error control) while preserving decent statistical power. The paper provides both theoretical guarantees and empirical evaluations on synthetic and real-world datasets (public-health insurance coverage and hospital readmissions).

**Claims And Evidence:**

- SHIFT detects subgroups with large performance decay due to distribution shifts, specifically distinguishing whether the shift is driven by covariates or by the outcome distributions.
- SHIFT can then explain decay with sparse variable subsets, checking if these smaller shifts can plausibly account for the large performance drop in the discovered subgroups.
- Tests have valid Type I error control and good power asymptotically, meaning SHIFT will not flag a nonexistent shift too often, and will detect real shifts with high probability given enough data.
- SHIFT helps practitioners mitigate shifts more effectively than blanket retraining, by highlighting targeted fixes that resolve problems within critical subgroups without negatively impacting performance elsewhere.

**Essential References Not Discussed:**

There are no additional related works that I believe need to be cited.

**Experimental Designs Or Analyses:**

Yes, I went through Section 5 in detail. My main concern with the experimental design is the choice of case study datasets. It’s unclear why these datasets were chosen and what underlies these datasets that we should expect their to be distribution shifts to test for. Furthermore, the readmission case study is unclear as to who the subgroups are.

**Methods And Evaluation Criteria:**

The proposed methods make sense for the problem at hand given the decomposition of distribution shift into these constituent components at the covariate and outcome level. Meanwhile the evaluation criteria seems applicable for driving application but less clear how this fits into the benchmarks used in other related works. For example, when comparing TE-VIM to SHIFT, the real world datasets are not those used in the original paper. The synthetic experiment does a good job of making it clear why and when SHIFT will outperform methods tailored to each type of shift. It is much less clear why the real-world case studies are relevant in the context of the prior work.

**Other Comments Or Suggestions:**

None.

**Other Strengths And Weaknesses:**

Strengths:

- Clear formalization of the testing framework and its relation to existing literature
- Test unifies two areas of research producing a more useful test for practitioners

Weaknesses:

- Applied case studies section lacks clarity and unconvincing of the methods utility in settings outside of the synthetic setups
- Synthetic setups could be more tailored to demonstrate why SHIFT outperforms prior methods
- Theory is derived to show bounds on the Type I error but this isn’t measured in the empirical results

**Questions For Authors:**

Below is a list of questions for which I would engage in a discussion that would potentially lead to increasing my score to a weak accept or accept:

1. Can you describe why this method outperforms other methods on the respective two tasks presented? If the claim is not the this method outperforms those but its utility is in being able to use it for both tasks at once then how should I think about that efficiency gain? It’s unclear to me how much more efficient it would be than just using two separate tests – if I knew which ones to use of course.
2. I would like a much more detailed description and understanding of the case studies. What background underlies the choice in subgroup? Do we expect their to shifts in their distributions? What are the subgroups for readmission?
3. If you fix the issues in the encounters feature do you actually see the accuracy gaps close?
4. How does this method work when we scale the size of subgroups that experience a drop in accuracy? Given the epsilon parameter defines the size of the subgroup there are an arbitrary number of subgroups one can construct of that size. I assume there is a breaking point at which the method fails to detect if that epsilon is too small.
5. Finally, I’m curious about how the dimensionality of the problem would factor into your theoretical and empirical results. Currently, I don’t see any dimensionality issues in the theory which makes sense given the nature of the test. Empirically though, the experiments are all in quite low dimensional settings. I’d be interested in understanding at what X variable subsets the test struggles.

**Relation To Broader Scientific Literature:**

The key contribution of this paper in relation to the existing literature on distribution shift detection and heterogeneous subgroup performance is the formalization of the sources of heterogeneity and a universal test to distinguish both issues. Much of the literature has been disparate in aiming to develop statistical tests for either issue. To my knowledge, this is the first test to present this hierarchical test.

**Theoretical Claims:**

I briefly skimmed the the theoretical results in Appendix C. Primarily focusing Theorem C.2.

---

> ### Author Rebuttal · Authors · 2025-04-01
>
> We thank the reviewer for carefully reading the work and for appreciating its practical relevance. Indeed, heterogeneity in ML performance is a major safety concern in high-risk applications and there is no unified test to identify the sources of heterogeneity.
>
> ---
>
> **Why SHIFT outperforms other methods** There are two reasons why SHIFT is better. First, SHIFT is much more targeted: whereas methods like MMD and KCI try to detect *any* shift which dilutes their power to find the more relevant shifts, SHIFT is only interested in finding *subgroups* with a performance drop and the subgroups have to satisfy minimum size and magnitude requirements. Second, SHIFT uses ML techniques to perform the test, whereas the comparators are kernel-based and do not scale well to high dimensions.
>
> Another advantage of a unified framework is that the test results across the aggregate and detailed levels will not conflict with each other. The current approach in the literature is to use a mix of different tests for the two levels, which may lead to conflicting results that are challenging to reconcile since the tests often have different null hypotheses.
>
> ---
>
> **Background of case studies** We chose the case studies to mirror the real-world application of the framework. They consist of settings where covariate or outcome shifts impact performance and domain experts do not know which shifts are detrimental. Such settings are highly prevalent in healthcare where ML performance varies widely across hospitals and time. We do not use the same data as the TE-VIM baseline since it is for causal inference and only assesses outcome shifts. We will update the paper with the following background.
>
> The first case study is based on a systematic analysis in Liu et al. 2023 that analyzed performance drops of an algorithm for predicting insurance coverage across different US states in the ACS dataset. Among many state pairs, Liu et al. primarily found a large decay when transfering the algorithm from Nebraska to Louisiana. We decided to dive deeper into this analysis by identifying which subgroups were affected and why. SHIFT detected that people who are unemployed or whose parents are not in the labor force experience a large decay (Fig 3c). Since health insurance coverage is tied to employment in the US, and insurance rates and incomes differ between the states, such a decay is expected.
>
> The readmission case study analyzes an algorithm to predict readmission that is trained on a well-resourced academic hospital and applied to a safety-net hospital. Since safety-net hospitals serve patients regardless of their ability to pay, their populations are quite different. SHIFT detected that patients with many emergency encounters experience a large decay (Fig 3d), which is expected because safety-net hospital patients seek care from emergency departments for very different reasons than at academic hospitals. Thus, SHIFT helps detect subgroups in realistic benchmarks.
>
> ---
>
> **Fix issues in encounters feature** We processed the readmission data again to correct the encounters feature. After correction, covariate shifts no longer lead to a significant subgroup-level accuracy drop (p-value goes from 0.00 to 0.69). Thus, SHIFT has helped bridge the accuracy gap.
>
> ---
>
> **Effect of scaling subgroup size** As the prevalence of the subgroup experiencing performance decay decreases, the power of SHIFT decreases. If the prevalence of the subgroup goes so low to be below the specified minimum threshold in SHIFT, the null hypothesis would be true and this tiny subgroup would no longer be of interest. Thus the "breaking point" of SHIFT is the specified minimum threshold, but this is *by design*. Other tests also decrease in power as the subgroup experiencing the shift decreases in prevalence. But existing tests set the minimum threshold to zero, stating that all shifts are of practical interest and yet have limited power to detect them. For empirical results of SHIFT for small subgroups, please refer to Fig 6, 7 in Appendix.
>
> ---
>
> **Effect of dimensionality** The theoretical results primarily rest on one's ability to estimate the nuisance functions at a sufficiently fast rate. When dimensionality increases, estimation rates for nuisance functions tend to slow down, though it may still be sufficiently fast if these functions are sparse [see e.g. 1,2]. To test SHIFT empirically, we applied our method to a text classification problem with 768-dimensional embeddings and was able to detect shifts. For the revised manuscript, we will include more validation of SHIFT in higher-dimensional datasets.
>
> ---
>
> **Type I error** Please refer to Fig 6, 7 in Appendix, where we confirm that Type I error of SHIFT is controlled.
>
> We hope that our responses clarify the concerns raised.
>
> ---
>
> [1] Wager et al. Adaptive concentration of regression trees... arXiv 2015
>
> [2] Belloni et al. $\ell_1$-penalized quantile regression... Ann Statist 2011

---

### Decision · Program_Chairs · 2025-05-01

**Decision:**

Accept (poster)

**Comment:**

This work studies subgroup performance degradation through their new method SHIFT. The reviews were largely positive (2,3,3,3) with reviewers noting the strong theoretical and empirical claims. I want to commend the authors for their detailed and clear responses to reviewer concerns, including the relevance of the real-world case studies (GEzi), effects on high-dimensionality (GEzi), and runtime and compute costs (GEzi, buFa, g6ej). The most negative review came from JHtq, who did not engage with the authors following their detailed responses and whose review was quite short and vague.

The contribution appears sound and well-motivated. As a result, I recommend accept.